# Single-molecule tracking of Nodal and Lefty in live zebrafish embryos supports hindered diffusion model

Timo Kuhn[1], Amit N. Landge [2,5], David Mörsdorf [3,4,5], Jonas Coßmann[1], Johanna Gerstenecker[1], Daniel Čapek [2], Patrick Müller [2,3] & J. Christof M. Gebhardt [1]

The hindered diffusion model postulates that the movement of a signaling molecule through an embryo is affected by tissue geometry and binding-mediated hindrance, but these effects have not been directly demonstrated in vivo. Here, we visualize extracellular movement and binding of individual molecules of the activator-inhibitor signaling pair Nodal and Lefty in live developing zebrafish embryos using reflected light-sheet microscopy. We observe that diffusion coefficients of molecules are high in extracellular cavities, whereas mobility is reduced and bound fractions are high within cell-cell interfaces. Counterintuitively, molecules nevertheless accumulate in cavities, which we attribute to the geometry of the extracellular space by agent-based simulations. We further find that Nodal has a larger bound fraction than Lefty and shows a binding time of tens of seconds. Together, our measurements and simulations provide direct support for the hindered diffusion model and yield insights into the nanometer-to-micrometer-scale mechanisms that lead to macroscopic signal dispersal.

The development of an embryo from a single cell to a complex organism is coordinated by cellular communication via signaling molecules called morphogens. Morphogens are produced in localized sources, from which they spread to form concentration gradients. Target cells along a morphogen gradient perceive different amounts and durations of morphogen signaling and respond by switching on different cell fate programs. By coupling molecular concentrations to distributions in space, morphogens can therefore provide positional information to orchestrate tissue patterning[1].

The range of a morphogen gradient needs to span multiple cell diameters from the source in order to provide positional information. While special transport mechanisms – for instance along cell extensions – are important in certain developmental contexts[2–6], the most prominent theory to explain the establishment of a morphogen gradient is the synthesis-diffusion-clearance model[7–13]. In this model, morphogens are produced in a localized source, from which they spread into neighboring tissues by diffusion. The length-scale of the gradient is determined by morphogen clearance – degradation or cellular uptake – as well as the morphogen's diffusivity. While the free diffusivity of a morphogen is a biophysical property that can be influenced by factors in the tissue environment such as temperature and viscosity, the hindered diffusion model postulates that the effective diffusivity of a molecule can be further influenced by transient binding interactions[2,7]. Indeed, there are numerous reports demonstrating direct binding of morphogens to intra- and extracellular molecules such as receptors[14,15], collagen[16] and heparin sulfate proteoglycans[7,13,17–22] that can modulate the shape of a morphogen gradient, but it remains unclear whether binding truly affects

[1]Institute of Biophysics, Ulm University, Albert-Einstein-Allee 11, 89081 Ulm, Germany. [2]University of Konstanz, Universitätsstraße 10, 78464 Konstanz, Germany. [3]Friedrich Miescher Laboratory of the Max Planck Society, Max-Planck-Ring 9, 72076 Tübingen, Germany. [4]University of Vienna, Department of Neurosciences and Developmental Biology, Djerassiplatz 1, 1030 Vienna, Austria. [5]These authors contributed equally: Amit N. Landge, David Mörsdorf. ✉e-mail: patrick.mueller@uni-konstanz.de; christof.gebhardt@uni-ulm.de

morphogen diffusivity or rather retention, uptake and stability. Beyond flat tissue culture systems[23], the tenets of the hindered diffusion model – i) free diffusion far away from cell surfaces, ii) hindered diffusion due to the tissue architecture, iii) further reduction due to binding[2,7] – have not been directly demonstrated for any morphogen in an in vivo tissue context.

The secreted TGF-β superfamily ligands Nodal and Lefty are prime examples of an activator-inhibitor morphogen pair whose different signaling ranges have been postulated to arise from differential hindrance[2,24,25]. This system has been best characterized in zebrafish embryos, where the Nodal signaling proteins Squint and Cyclops are produced in the marginal zone and induce the formation of mesoderm and endoderm during early development, beginning around 4 h post-fertilization (hpf)[9,26–30]. Nodal signaling is antagonized by secreted Leftys[27,31–34], which inhibit Nodals from binding to their receptors[35,36].

Hindered diffusion has been proposed to result in the formation of Nodal and Lefty concentration gradients[2,25] where Cyclops has an ultra-short range of only a few micrometers, Squint has a short-to-mid range, Lefty1 acts at a long range, and Lefty2 has an ultra-long range leading to a nearly uniform distribution throughout the embryo[25]. Previous observations of the Nodal/Lefty system are consistent with the hindered diffusion model. First, free diffusion coefficients measured by fluorescence correlation spectroscopy (FCS) in a diffraction-limited spot far away from cell surfaces yielded similar local diffusion coefficients on a sub-micrometer scale for zebrafish Nodals and Leftys[2,14,25]. Second, effective diffusion coefficients on a tissue level across a cube of approximately $8 \times 8 \times 8$ cells measured by fluorescence recovery after photobleaching (FRAP) were found to be much lower for Nodals than for Leftys (Cyclops < Squint < Lefty1 < Lefty2)[25,37–39]. Third, it has been shown that Nodals bind their receptors with nanomolar affinity[14], and manipulating the levels of the co-receptor Oep modulated the Nodal signaling range and distribution[40,41]. However, it remains unclear whether and how such treatments affect Nodal and Lefty movement, how effective diffusivity through a tissue emerges from interactions at the molecular scale, how binding on the cell surface contributes to Nodal and Lefty movement, and how tissue geometry affects morphogen spreading.

Here, we present single-molecule imaging and tracking of HaloTag-tagged fluorescent Cyclops, Squint, Lefty1 and Lefty2 in the extracellular environment of live developing zebrafish embryos. We monitored the movement of these morphogens on the nanoscale and observed a major influence of the local extracellular architecture on the diffusion properties. We found that molecules moving in extracellular cavities between cells were predominantly diffusing freely. In contrast, we observed hindered diffusion within cell-cell interfaces with larger bound fractions of Nodal molecules compared to Lefty. Time-lapse microscopy enabled us to observe individual binding events of tens of seconds for Cyclops and Squint. We developed an agent-based model of single-molecule movements and found a major contribution of tissue architecture, receptor levels and affinity on morphogen distributions. Overall, our single-molecule fluorescence measurements directly support a model of hindered diffusion for Nodal and Lefty, where Nodals – but not Leftys – are transiently trapped on the cell surface, explaining their short action range.

## Results

### Single-molecule imaging of HaloTag-labeled morphogens in live zebrafish embryos

To observe the movement of individual morphogens, we used a reflected light-sheet microscope (RLSM), which is ideally suited to image single molecules in live embryos[42,43]. In order to visualize Nodals and Leftys we fused them to HaloTags. We inserted the HaloTag between the pro- and mature domains of Cyclops and Squint and added them to the C-termini of Lefty1 and Lefty2, generating active and properly localized proteins analogous to previous approaches[25] (Fig. 1a, Supplementary Fig. 1, Materials and Methods). The HaloTag allows precise titration of the amount of fluorescence label, ensuring low densities of labeled molecules in every frame over the entire measurement period (Supplementary Fig. 2). To visualize single molecules, we injected embryos at the one-cell stage with only 1-2 pg of each mRNA (Fig. 1b, Material and Methods), 30 times less than what has been used for the assessment of effective diffusivities in FRAP experiments[25] and 60 times less than what is required to induce a full body axis in zebrafish[37]. In addition, we co-injected mRNA encoding membrane-targeted green fluorescent protein[44] (memGFP) to visualize cell outlines. After injection, embryos were incubated in JF549[45] dye solution to covalently label the HaloTag fusion protein. Subsequently, we extensively washed the embryos to remove unbound dye (Fig. 1b, Material and Methods).

We started to image the embryos with our RLSM setup at the end of the 128-cell stage (Fig. 1c) and continued the measurements up to sphere stage, shortly before gastrulation[46]. All fluorophores detected in each frame were used to track molecules in time using the software TrackIt[47] (Material and Methods). Given the compartmentalization of the embryonic tissue into intra- and extracellular regions, we performed our tracking analysis in separate sub-regions. To automatically identify extracellular regions, we analyzed memGFP images by training a convolutional neural network (CNN)[48] with a manually annotated data set (Fig. 1d, Material and Methods). The intra- and extracellular masks classified in this manner were visually inspected and manually corrected. On average 33% of a prediction mask was truncated and subsequently 8% manually corrected. The early blastoderm comprises loosely packed cells[49], subdividing the extracellular space into regions of close cell-cell contacts and large intercellular cavities where cell contacts are missing. We therefore further manually classified the extracellular space into interface and cavity regions and performed our single-molecule analysis separately in those regions (Fig. 1d, Material and Methods).

### Nodals have similar diffusion coefficients but higher immobile fractions compared to Leftys

We first characterized the mobility of Nodals and Leftys in interfaces and cavities of the extracellular space by acquiring continuous movies of each morphogen at a rate of 85 frames per second (Supplementary Movie 1). When the single-molecule positions were integrated over all frames, the distributions of Nodals and Leftys well resembled the known localizations[25]: Cyclops was largely found in puncta, Squint in both puncta and diffusely, whereas Lefty1 and Lefty2 nearly uniformly occupied the extracellular space (Fig. 2a, Supplementary Fig. 1, Supplementary Movies 2–5). Interestingly, all secreted molecules were more likely to be found in cavities than in interfaces based on the ratio of localization densities as a measure of the probability to encounter a morphogen in one of the two extracellular compartments (Fig. 2b), and we found an increase in the localization density ratio commensurate with the morphogens' effective global diffusivities[25] (Cyclops: 1.53-fold, Squint: 2.66-fold, Lefty1: 3.83-fold, Lefty2: 4.03-fold, secHalo: 4.20-fold).

We then sorted the distances between consecutive fluorophore localizations within a track (jump distances) into histograms (Fig. 3a). These distances constitute two-dimensional projections of the three-dimensional tracks detected within the depth of focus of the objective. In both interfaces and cavities, Cyclops, and to a lesser degree Squint, showed a larger probability of short jump distances (<0.3 μm) than Lefty1 and Lefty2, indicating reduced mobility of Nodals. In contrast, in cavities, both Nodals and Leftys exhibited a higher probability of long jump distances – and hence higher mobility – compared to interfaces. We quantified the mobility of morphogens by analyzing the corresponding cumulative distribution of jump distances[50] (Fig. 3b). In our analysis, we did not correct for bias from the projection of tracks or

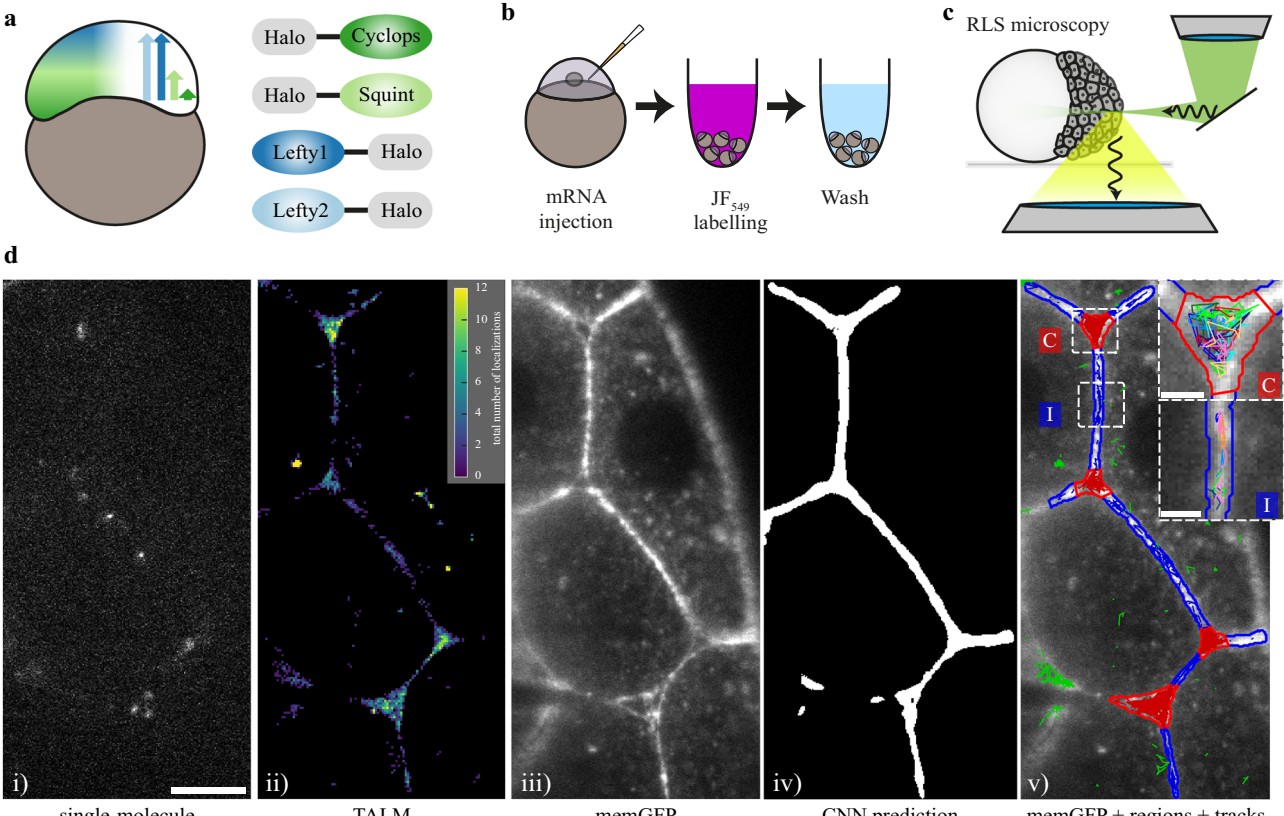

**Fig. 1 | Single-molecule imaging of HaloTag-labeled Nodal and Lefty in live zebrafish embryos using RLSM. a** Schematic of the presumed concentration gradients of Nodal (dark and light green) and Lefty (dark and light blue) in early zebrafish embryos (left) and sketch of HaloTag fusion proteins of mature Cyclops, mature Squint, Lefty1 and Lefty2 (right). **b** Schematic of the labeling workflow: mRNA encoding the fusion proteins was injected at the one-cell stage. Embryos were incubated in medium containing the HaloTag JF-549 ligand to covalently label the HaloTag. Excess dye was removed in washing steps. **c** Sketch of a zebrafish embryo imaged with a reflected light-sheet microscope. **d** Workflow of single-molecule imaging and image segmentation: i) signal of single Lefty2-HaloTag molecules at 561 nm laser illumination for 10 ms; ii) tracking-and-localization-microscopy (TALM) image showing the total number of localizations over 1000 frames or 11.7 s in each 2 × 2 pixel bin; iii) memGFP signal at 488 nm laser illumination averaged over 10 × 10 ms frames, outlining cellular membranes; iv) region of interest (ROI) mask of the extracellular space predicted by a convolutional neural net (CNN) based on the memGFP signal; v) overlay of the memGFP signal, the manually curated ROI separating cell-cell-interfaces (blue, I) and extracellular cavities (red, C); tracks assigned to interfaces (blue), cavities (red) and not assigned (green) are also shown. Scale bar: 10 μm. Insets in v): zoom of the indicated interface and cavity including an example set of tracks. Scale bar: 2 μm.

out-of-focus movement, which similarly applies to all morphogens and therefore does not alter their relative behavior. A three-component Brownian diffusion model best described the data (Supplementary Fig. 3), yielding the diffusion coefficients $D_{1,2,3}$ of slow, intermediate and fast diffusion and their relative amplitudes $A_{1,2,3}$[43,51–53]. The slow diffusion component likely originates from immobile or slowly moving morphogens, although the uncertainty of localizing single molecules may also be a contributor. The intermediate and fast diffusion components together approximate anomalous diffusion, which has been observed in many systems[50,52,54–58]. For the morphogens, anomalous diffusion likely arises from diffusion in the spatially restricted, highly complex and heterogeneous environment of the extracellular space.

We found that in both interfaces and cavities, the diffusion coefficients of intermediate ($D_{2,i}$ in interfaces ≈ 1.2–3.0 μm²s⁻¹, $D_{2,c}$ in cavities ≈ 4.8–6.7 μm²s⁻¹) and fast ($D_{3,i}$ ≈ 16–17 μm²s⁻¹, $D_{3,c}$ ≈ 26–30 μm²s⁻¹) diffusion were largely comparable between all four morphogens, yet overall higher in cavities (Fig. 3c, Supplementary Table 1 and Supplementary Table 2). The fast diffusion coefficients in cavities were comparable to those measured previously with FCS, which were reported in a range between ~30 μm²s⁻¹ and ~60 μm²s⁻¹ for Nodal and Lefty[2,14,25]. Interestingly, Cyclops showed lower diffusion coefficients in the immobile and intermediate diffusion classes, in agreement with its punctate localization pattern (Supplementary Fig. 1) and potentially indicating higher confinement of this morphogen. Diffusion of the

HaloTag alone (sec-Halo) was faster than any morphogen in both interfaces and cavities in accord with its smaller size, whereas the HaloTag fused to GFP showed diffusion coefficients similar to those of the morphogens (Supplementary Fig. 4, Supplementary Table 1 and Supplementary Table 2).

The fraction of immobile molecules in interfaces and cavities was considerably larger for Cyclops ($A_{1,i}$ in interfaces 44%, $A_{1,c}$ in cavities 22%) and Squint ($A_{1,i}$ ≈ 35%, $A_{1,c}$ ≈ 9%) than for Lefty1 ($A_{1,i}$ ≈ 21%, $A_{1,c}$ ≈ 5%) and Lefty2 ($A_{1,i}$ ≈ 24%, $A_{1,c}$ ≈ 6%) (Fig. 3d), reflecting the higher probability of short jump distances for Nodals (Fig. 3a). Correspondingly, while the fraction of molecules with intermediate diffusivity was similar for Nodals and Leftys, the fraction of fast-diffusing molecules in interfaces and cavities was larger for Lefty1 ($A_{3,i}$ ≈ 40%, $A_{3,c}$ ≈ 64%) and Lefty2 ($A_{3,i}$ ≈ 37%, $A_{3,c}$ ≈ 61%) than for Cyclops ($A_{3,i}$ ≈ 17%, $A_{3,c}$ ≈ 48%). The fraction of fast-diffusing Squint molecules was lower than that of Leftys in interfaces, but comparable to the fraction of fast-diffusing Leftys in cavities ($A_{3,i}$ ≈ 31%, $A_{3,c}$ ≈ 63%) (Fig. 3d, Supplementary Table 1 and Supplementary Table 2). Taken together, our analysis of the diffusion data confirms that the fast diffusion coefficients of Nodals and Leftys are comparable[2,14,25] and suggests that the differential mobility of Nodal and Lefty reported previously[2,25,37,39] originates from a higher retention of Nodal in an immobile state. This retention is more efficient in interfaces, where the fractions of immobile molecules are larger and diffusion is slower than in cavities.

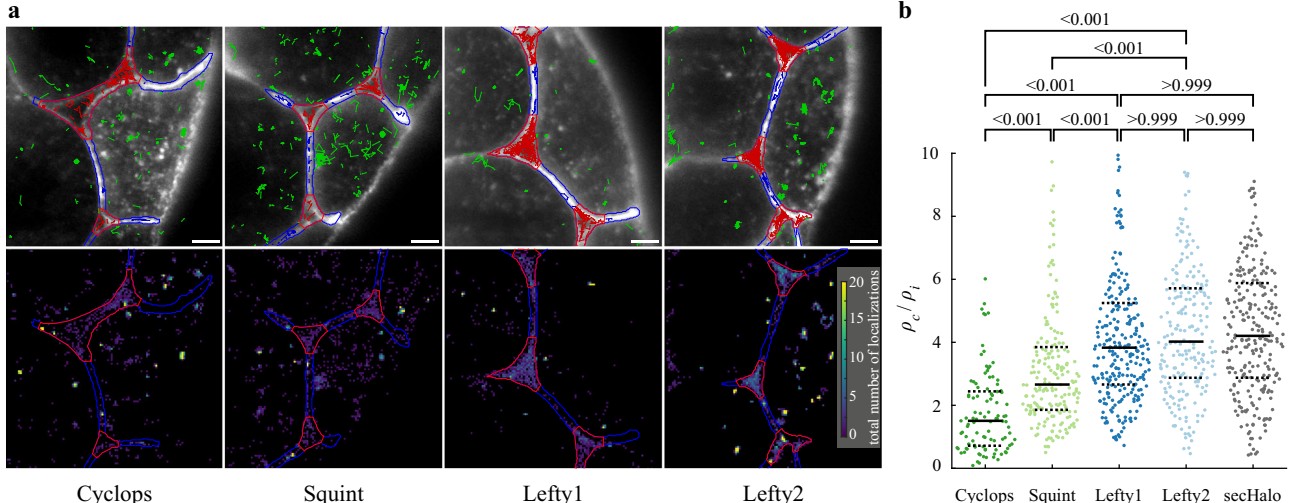

**Fig. 2 | Localization of morphogens in cell-cell interfaces and extracellular cavities. a** Top: Overlay of the memGFP signal, the manually curated ROI separating cell-cell-interfaces (blue, I) and extracellular cavities (red, C); tracks assigned to interfaces (blue), cavities (red) and not assigned (green) are also shown. Bottom: Tracking-and-localization-microscopy (TALM) image showing the total number of localizations over 1000 frames or 11.7 s in each 2 × 2 pixel bin; scale bar: 5 μm. **b** Ratios of localization densities in cavities and interfaces calculated for each movie

(for statistics see Supplementary Table 5). Values above 10 are not displayed (3.4% of all movies). *p*-values (Cyclops - Squint, Cyclops - Lefty1, Cyclops - Lefty2, Squint - Lefty1, Squint - Lefty2: <0.001; Lefty1 - Lefty2, Lefty1 - secHalo, Lefty2 -secHalo: >0.999) were calculated using the Kruskal-Wallis-Test. Solid black lines indicate the median values, dashed black lines the 0.25 and 0.75 quantiles. Source data are provided as a Source Data file for Fig. 2b.

To explore the differential diffusion properties in interfaces and cavities, we calculated the angles within a track spanned by three consecutive localizations[59,60]. We only considered angles where the two jumps making up the angle covered a minimum distance of 1 px (166 nm), much larger than the localization error. In interfaces, both Nodal and Lefty showed an anisotropic angle distribution with a high probability to continue or reverse the previous direction (Fig. 3e). In contrast, the angle distribution was more isotropic in cavities. The angle distributions of both interfaces and cavities exhibited a prominent contribution of reverse motion (Supplementary Fig. 5a). In interfaces, jumps were mostly oriented along the direction of interface borders and the probability for a subsequent jump to go into the reverse direction strongly increased with the angle of the preceding jump to the interface border (Supplementary Fig. 5b, c, top). These effects were less prominent for cavities (Supplementary Fig. 5b, c, bottom). Overall, these observations reflect hindered diffusion in limited space, which is more restrictive in interfaces than in cavities.

### Nodals bind in the extracellular space with retention times of ten to twenty seconds

To test the idea that Nodals are trapped in cell-cell interfaces, we next characterized the residence times of Cyclops and Squint in the bound state. We used time-lapse imaging, where two images are separated by dark times of different duration (Fig. 4a, Material and Methods). With this illumination scheme, it is possible to increase the measurable range of binding times and to resolve several photobleaching-corrected dissociation rate constants[47,61–63]. We chose frame cycle times of 11.7 ms, 58 ms, 199 ms and 1006 ms and were thus able to observe binding events of tens of seconds along the membrane for both Cyclops and Squint (Fig. 4).

We identified bound molecules in interfaces and cavities using a small tracking radius in combination with a minimum number of survived frames in the nearest neighbor algorithm[47] (Material and Methods). We then collected the durations of binding events for each time-lapse condition in survival-time distributions (Fig. 4c, Material and Methods). The distributions extended to longer durations for Cyclops than for Squint, indicating longer binding times for Cyclops. For the longest time-lapse condition, where photobleaching is not limiting, few binding events survived throughout the whole acquisition time (5% for Cyclops, 1.4% for Squint). Thus, our analysis will slightly underestimate the binding times. Lefty1 and Lefty2 exhibited much-reduced occurrences and durations of binding events in movies of 11.7 ms frame cycle time, comparable to those of the HaloTag alone (Supplementary Fig. 6a). Together with the low bound fractions obtained from the diffusion analysis, this indicates that binding in the extracellular space has a minor influence to the overall diffusion properties of Leftys, and we therefore refrained from quantifying their binding times. For Cyclops and Squint, we analyzed the survival-time distributions with our genuine rate identification (GRID) tool, which can reveal spectra of dissociation rates from fluorescence survival-time distributions by solving the inverse Laplace transformation[61] (Material and Methods). We obtained four dissociation rate clusters for both Cyclops and Squint (Fig. 4d), from which the inversely correlated binding times can be calculated. The longest binding time, corresponding to the slowest dissociation rate cluster, was $16.2 \pm 2.6$ s (mean ± s.d. of resampled spectrum), comprising $50.0 \pm 4.1\%$ (mean ± s.d. of resampled spectrum) of bound molecules for Cyclops. For Squint, we found shorter binding times of $11.0 \pm 2.2$ s comprising $28.9 \pm 4.8\%$ of bound molecules (Fig. 4e), consistent with the larger effective diffusion coefficient, the less punctate distribution compared to Cyclops[25] (Supplementary Fig. 1), and in good agreement with the previous dissociation rate predictions of 18 s for Cyclops and 4 s for Squint[25].

Some of the Cyclops and Squint molecules that we identified as bound showed slow diffusive motion along the membrane (Fig. 4b, Supplementary Movies 6, 7). This observation is in agreement with a fraction of slowly diffusing morphogens obtained in the analysis of molecular jump distances (Fig. 3c, d). Such motion might correspond to the diffusion of morphogen-receptor complexes within the membrane. To test this idea, we quantified the diffusion coefficients of bound morphogens by analyzing the mean-squared displacement (MSD) of bound tracks for different time intervals (Supplementary Fig. 7, Supplementary Movies 6, 7, Material and Methods). The majority of diffusion coefficients of both Cyclops and Squint was below 0.5 μm²s⁻¹ (Fig. 4f), indeed similar to previous quantifications of receptor diffusion in membranes[64,65]. Bound Squint molecules exhibited a higher tendency to diffuse along the membrane than bound

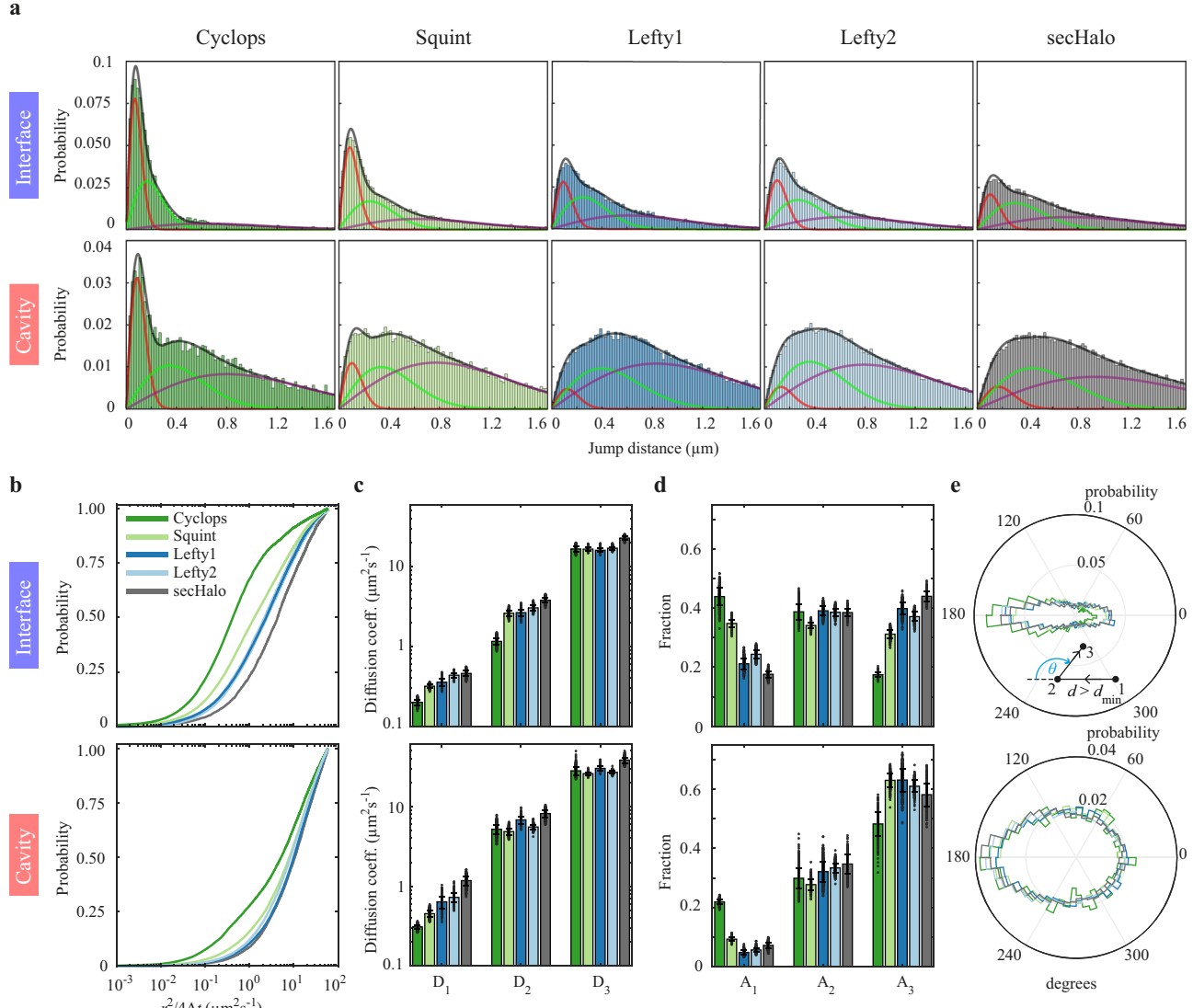

**Fig. 3 | Mobility of morphogens in cell-cell interfaces and extracellular cavities.** **a** Distribution of jump distances within single-molecule tracks in interfaces and cavities for the indicated morphogen and sec-Halo. Lines represent a three-component diffusion model (black) and the individual components (red, green, purple). **b** Cumulative distributions of jump distances. **c**) Diffusion coefficients and (**d**) fractions of the three-component diffusion model (Supplementary Table 1, 2). Data are presented as mean values ± s.d. from 500 resamplings with randomly selected 50% of the data. **e** Distribution of angles between two consecutive track segments. For full experimental statistics see Supplementary Table 5. Source data are provided as a Source Data file.

Cyclops molecules, which were more often confined to a small area, again consistent with the larger effective diffusion coefficient and less punctate distribution compared to Cyclops[25] (Supplementary Fig. 1).

## Overexpression of oep increases the fraction of immobile Squint molecules

While our single-molecule imaging approach enables us to classify bound molecules by their time they spend within a certain area, the origin of binding of individual tracks remains obscure. Binding might reflect both specific interactions with receptors or unspecific retention by the tortuous environment. Nodals are well known to bind to the EGF-CFC co-receptor Oep[36,66], which is essential for Nodal signaling[67]. Furthermore, the signaling range and distribution of Squint was shown to be extended in the absence of *oep*[40,41], but a direct effect of Oep on Nodal dispersal at the nanometer-to-micrometer scale has not yet been directly demonstrated. To test whether immobile Squint in our experiments was due to binding to Oep, we co-injected 0.3 pg, 3 pg and 30 pg of Oep-encoding mRNA together with the Squint-HaloTag

construct and compared the diffusion properties of Squint in conditions of *oep* overexpression with those of endogenous Oep levels. We found that the immobile fraction of Squint increased in interfaces and cavities, from 35% to 50% and from 9% to 25% respectively (Fig. 5b and Supplementary Fig. 8a), while the diffusion coefficients remained unaffected (Fig. 5a). Thus, our data show on a single-molecule level that Squint at least to some extent binds to Oep, which can directly hinder the diffusion of Nodal by transiently trapping the morphogen on the membrane.

## Clarifying origins of differential morphogen localization using agent-based modeling

Interestingly, we found a higher fraction of immobile molecules combined with slower diffusion in cell-cell interfaces compared to extracellular cavities. These findings would intuitively suggest that morphogens should accumulate in interfaces, not cavities. However – surprisingly, and in contrast to intuition – we found that all secreted molecules were more likely to be found in extracellular cavities rather than in cell-cell interfaces. Mathematical modeling can help reveal the

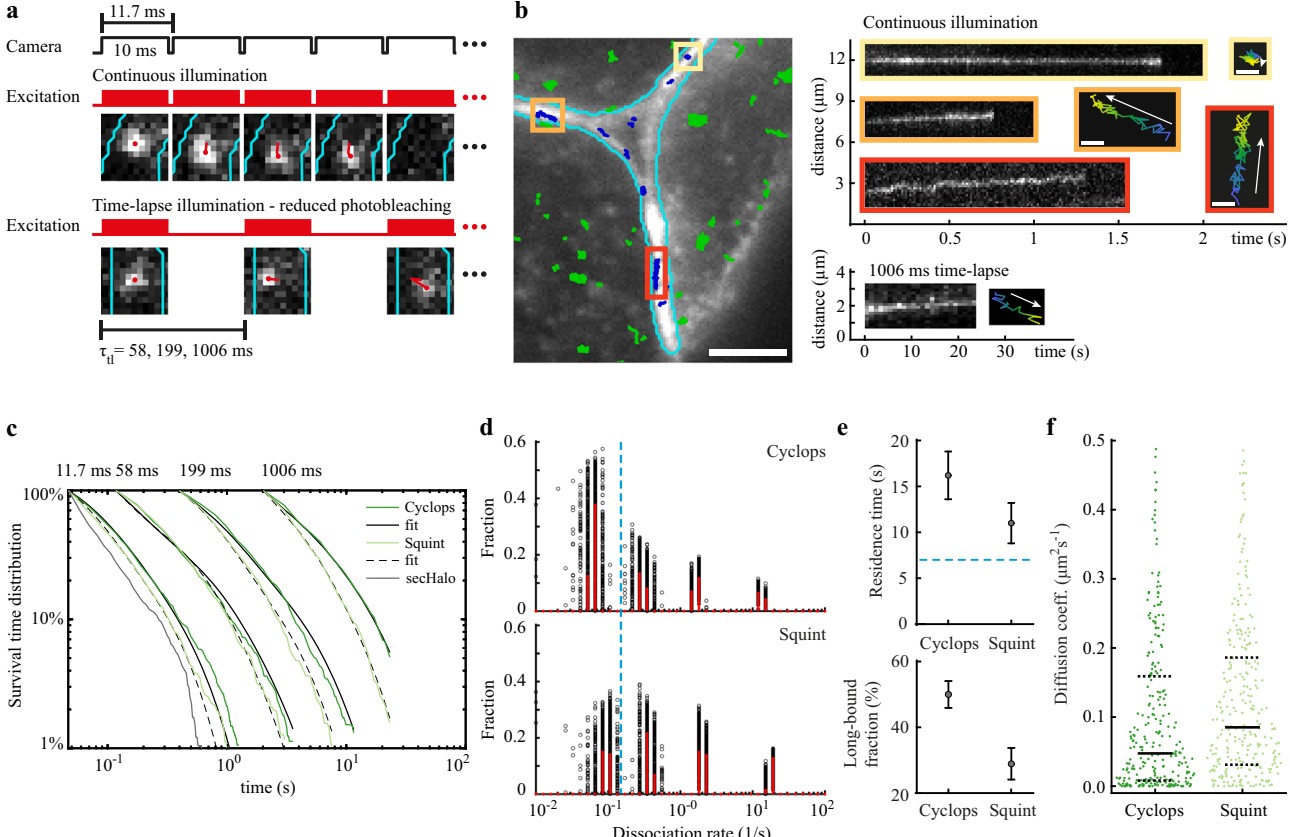

**Fig. 4 | Residence times of Cyclops and Squint in the extracellular space.**
**a** Overview of camera exposure and laser excitation patterns in time-lapse illumination experiments and representative images of single molecules overlaid with tracks (red) and the boundary of the extracellular region (cyan). **b** Left panel: Overlay of the memGFP signal (white) with the boundary of the extracellular region (cyan) and tracked Cyclops molecules from a movie with continuous illumination. Bound Cyclops molecules in the extracellular space shown in blue, intracellular molecules shown in green. Scale bar: 5 μm. Right panels: continuous illumination kymographs of the yellow, orange and red regions indicated in the left image and below a kymograph from a 1006 ms time-lapse movie as well as position plots of the tracks color-coded according to start (blue) and end (yellow) times. Scale bar: 0.3 μm. **c** Survival-time distributions of bound morphogens and sec-Halo in the time-lapse conditions indicated on top (dark and light green lines) and survival time functions obtained by GRID for Cyclops (solid black lines) and Squint (dashed black lines). **d** State spectra of dissociation rates of Cyclops and Squint obtained by GRID using all data (red bars) and 500 resampling runs with randomly selected 80% of data (black data points) as an error estimation of the spectra. The dashed blue line indicates the boundary for long-bound molecules of 0.15 s⁻¹. **e** Residence times of Cyclops and Squint (top panel) and the corresponding fraction (bottom panel) obtained from the slowest dissociation rate cluster of the state spectra. Error bars denote s.d. of the resampled spectra in **d**). The dashed blue line indicates the boundary for long-bound molecules of 7 s. **f** Diffusion coefficients obtained from fitting the first 10 points of a mean-squared displacement plot of bound Cyclops and Squint molecules recorded for at least 20 frames ($n_{tracks,cyc}$ = 283, $n_{tracks,squ}$ = 337). Values above 0.5 μm²s⁻¹ were discarded (11%). Solid black lines indicate the median values, dashed black lines are the 0.25 and 0.75 quantiles. For full experimental statistics see Supplementary Table 6. Source data are provided as a Source Data file for Fig. 4c, d, f.

origins of non-intuitive behaviors in biological systems[37,39,68]. We therefore devised a minimal model of single-molecule dispersal in order to test whether geometric constraints and binding might suffice to explain the cavity enrichment, or whether more complicated molecular mechanisms such as restricted entry control into interfaces have to be invoked.

To simulate single-molecule dispersal, we chose an agent-based model in a realistic zebrafish blastoderm geometry that directly relates to our experimental observations. We used an experimentally determined binary mask of extracellular space as two-dimensional simulation geometry (Fig. 6a). Single morphogens were simulated as "drunken sailors"[2] performing a random walk in the extracellular space. To simulate immobile and freely diffusing single molecules, we used jump sizes at each simulation step based on the measured diffusion coefficients for bound (0.5 μm²s⁻¹) and free (30 μm²s⁻¹) states (see Material and Methods). A single molecule became bound when it detected a receptor in proximity (≤ 20 nm). The simulated tracks for Nodals (Fig. 6a, blue track) and Leftys (Fig. 6a, red track) closely resembled the experimental observations. In particular, our simulation was able to recapitulate higher bound fractions in interfaces compared

to cavities, differential angle distributions in both compartments as well as higher localization density in cavities compared to interfaces (Fig. 6b, c, Supplementary Fig. 9a–f).

Next, we systematically varied key parameters that might modulate morphogen localization in the extracellular space. A parameter screen revealed that receptor density ($\sigma$), residence time ($\tau$), and width of the extracellular space in terms of extracellular fraction ($\eta$) are important determinants of extracellular molecule localization (Fig. 6b). As expected, strong binding ($\sigma > 0.5$ μm⁻¹ and $\tau > 6$ s) yielded localization in cell-cell interfaces (Fig. 6b, c and Supplementary Fig. 9a). Surprisingly, for low binding ($\sigma < 0.5$ μm⁻¹ and $\tau < 6$ s) extracellular molecules tended to localize in cavities. This tendency was augmented by decreasing the width of the extracellular space, suggesting an important role of tissue geometry for morphogen localization (Fig. 6b and Supplementary Fig. 9b). We verified the accumulation in cavities for multiple additional experimentally derived simulation geometries (Supplementary Fig. 9c). The narrow width of interfaces concentrated extracellular molecules to cell surfaces, thereby increasing the probability of interactions with receptors. This behavior resulted in a higher bound fraction in interfaces

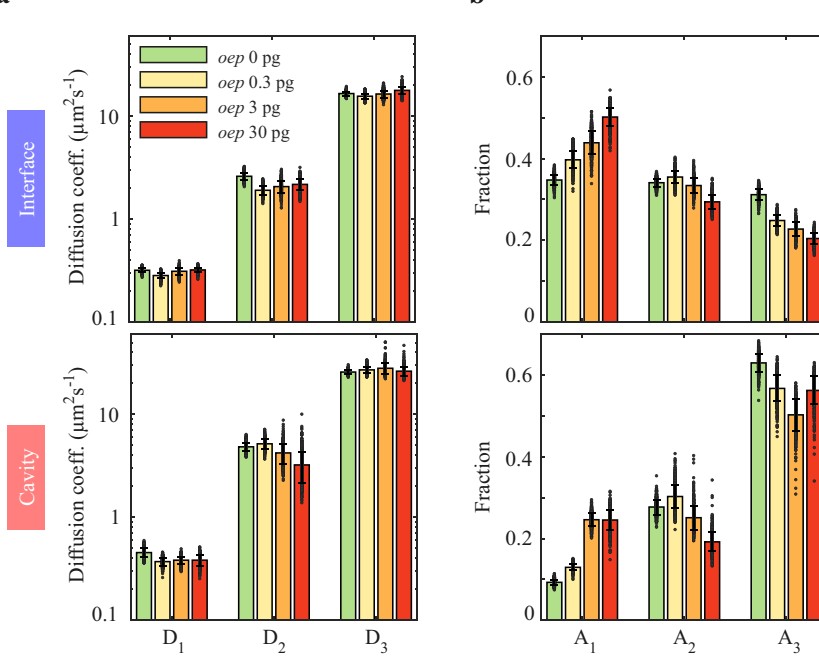

**Fig. 5 | Mobility of Squint decreases with overexpression of oep. a** Diffusion coefficients and (**b**) fractions of the three-component diffusion model in interfaces and cavities for Squint at different amounts of mRNA encoding Oep. Data are represented as mean values ± s.d. from 500 resamplings with randomly selected 50% of the data. For full experimental statistics see Supplementary Table 5. Source data are provided as a Source Data file.

compared to cavities despite similar receptor spacing in both compartments (Fig. 3d, Supplementary Fig. 9d, e).

Our simulations showed that the surprising enrichment of extracellular molecules in cavities can be explained purely by geometric constraints, and predicted that the diffusing molecules could be pushed out of this compartment into interfaces by increasing receptor density homogeneously in the tissue (Fig. 6c). To test this prediction, we measured the distribution of extracellular GFP in zebrafish embryos with different levels of membrane-tethered GFP-binding nanobodies[69]. Similar to our TALM findings (Figs. 1d, 2), secreted GFP was mainly distributed in cavities ($\rho_c/\rho_i > 2$) in case of no or low amount (25 pg) of injected nanobody mRNA. With high nanobody expression (100 pg), the GFP signal became more enriched in cell-cell interfaces ($\rho_c/\rho_i < 1$) (Fig. 6d and Supplementary Fig. 9g), in accordance with our simulations. Our results suggest that geometric constraints, such as the width of interfaces, bias morphogens to preferably localize in extracellular cavities. Strong binding to receptors can overcome this bias to increase morphogen localization in cell-cell interfaces.

Together, our single-molecule measurements and simulations provide strong support for the hindered diffusion model, in which differences in effective diffusivities between Nodals and Leftys are an emergent property arising from differential binding of morphogens in a compartmentalized extracellular environment.

## Discussion

Multiple mechanisms have been proposed to underlie the dispersal of morphogens for developmental patterning, from simple extracellular diffusion to repeated secretion and cellular uptake, filopodia-based distribution, and signal relay[2,9,13,14,30,70–77]. In particular, the hindered diffusion model postulating free diffusion intermitted by transient binding on cell surfaces has gained popularity[7]. Dispersal models have been inferred from observations of large averaged morphogen ensembles and bulk mobility measurements using techniques such as FCS or FRAP. However, the resulting data has to be carefully interpreted because bulk measurements may only provide indirect evidence for transport mechanisms[38,78–80]. To directly determine the

mechanisms of transport, single-molecule experiments are necessary, but these measurements have so far not been performed for any morphogen in a living embryo.

We performed single-molecule measurements of individual Nodal and Lefty morphogens in developing zebrafish embryos. Our results suggest that morphogens undergo fast diffusive motion in extracellular cavities, whereas diffusion is more constrained and slower in cell-cell interfaces. The coefficients of fast diffusion were similar for Nodals and Leftys. In contrast, Nodals, and in particular the ultra-short-range morphogen Cyclops, exhibited a larger fraction of molecules bound to the cell membrane than Leftys. Our direct single-molecule observations are consistent with previous inference from indirect bulk measurement techniques such as FCS and FRAP[2,14,25,38], and here we show both diffusion and reversible morphogen binding with single-molecule resolution in strong support of the influential hindered diffusion model.

Surprisingly, we observed an unexpected accumulation of morphogens in extracellular cavities, although binding interactions were more pronounced in cell-cell interfaces than in cavities. Using agent-based simulations of morphogen transport, we found that the architecture of the extracellular space with large cavities and narrow cell-cell interfaces favors heterogeneous distribution of morphogens and their accumulation in cavities. Our simulations predicted that binding to receptors would counteract this effect, and we validated this idea by measuring the distribution of extracellular GFP in zebrafish embryos with different levels of membrane-tethered artificial GFP-binding receptors[69]. In addition, stronger or more frequent binding, implemented by longer residence times or higher receptor densities, respectively, retained morphogens in cell-cell interfaces. Furthermore, the narrow width of interfaces contributed to enhanced binding by concentrating morphogens to cell surfaces.

The influence of tissue architecture on effective diffusion coefficients has been discussed in previous studies[2,14,38]: Numerical simulations and experiments using FCS and FRAP with secreted GFP inferred a reduction of effective diffusion compared to free diffusion by a factor of approximately two-fold[25,38], rationalizing the idea that

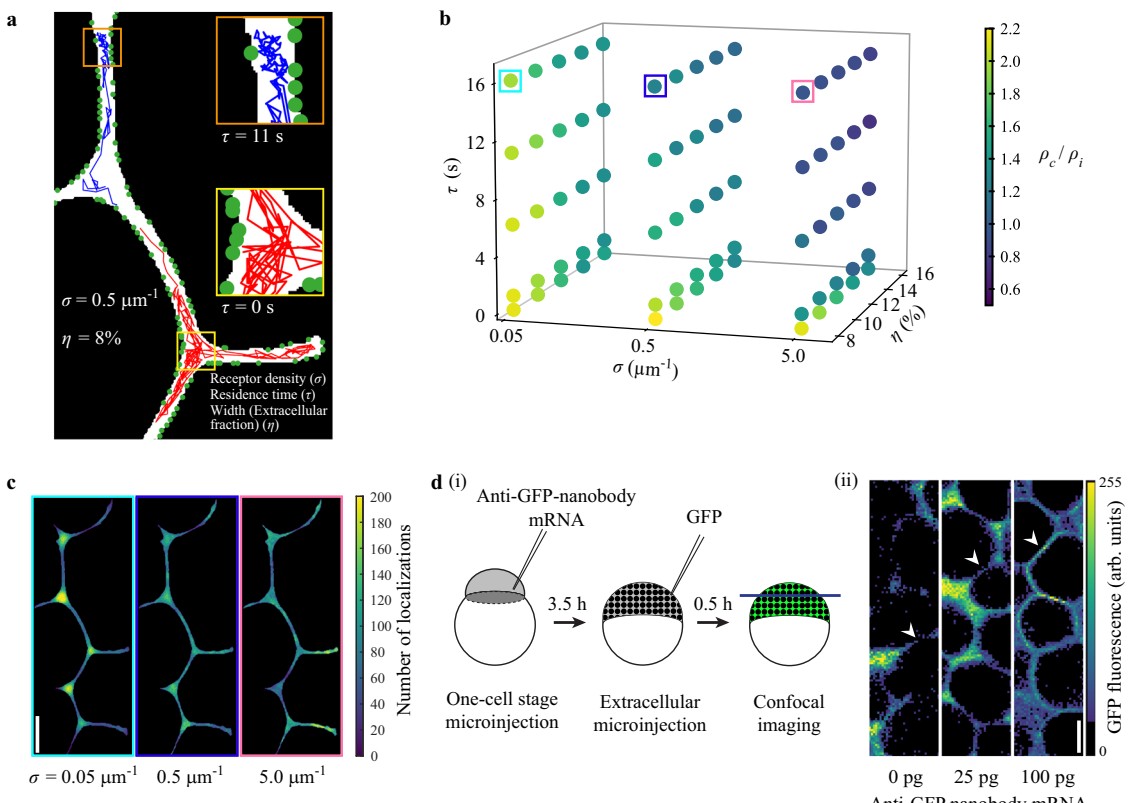

**Fig. 6 | An agent-based model reveals key parameters affecting morphogen behavior in the extracellular space. a** Illustration of the model with a section of the two-dimensional simulation geometry showing simulated tracks (200 steps or 2 s) for two morphogens with residence times $\tau$ of 11 s (blue track) and 0 s (red track), respectively. Green circles indicate receptors. The receptor density $\sigma$ is given as the number of receptors per μm of membrane length. The width of the extracellular region in terms of extracellular fraction is given by the parameter η. Morphogens are allowed to move in the extracellular space (white). A morphogen becomes 'immobile' or bound upon detecting a receptor in proximity and performs smaller jumps corresponding to the bound diffusion coefficient. **b** Three-dimensional scatter plot of simulated parameter combinations showing corresponding localization density ratios ($\rho_c/\rho_i$). Five residence times $\tau$ (0 s, 1 s, 6 s, 11 s, and 16 s), three receptor densities $\sigma$ (0.05 μm⁻¹, 0.5 μm⁻¹, and 5.0 μm⁻¹) and five extracellular fractions η (8%, 10%, 12%, 14%, and 16%) were simulated. **c** Localization density plots of simulated morphogens with increasing receptor densities $\sigma$ (0.05 μm⁻¹, 0.5 μm⁻¹, and 5.0 μm⁻¹; $\tau$ = 16 s, η = 8%). Frame colors correspond to colored boxes in **b**). **d** (i) Schematic depicting the experimental procedure to acquire confocal images of GFP localization in zebrafish embryos. (ii) Representative images of regions of interest used for GFP signal measurements. Cell-cell interface regions are indicated by arrowheads. Scale bars in c and **d** (ii) are 10 μm. *n* = 5 different regions of interest were measured per embryo and treatment; see Supplementary Fig. 9g for detailed statistical analysis. Source data are provided as a Source Data file for Fig. 6**b**.

secreted molecules have to bypass other cells compared to diffusion in free space. Our direct single-molecule measurements and simulations suggest a molecular mechanism of long-range morphogen spreading in a tortuous environment, where diffusion predominantly occurs in a network of extracellular cavities. This mechanism differs from the morphogen dispersal mode recently described in a two-dimensional human embryonic stem cell culture system, in which – unlike in the three-dimensional embryo context – Nodal molecules presumably cannot be retained in the extracellular space and are instead lost into the culturing medium[75]. Cavity accumulation in the loosely packed zebrafish blastoderm may allow for the rapid morphogen transport required to accomplish tissue patterning within the short, hour-long timescales during early embryogenesis. Dynamic changes in tissue geometry and cell numbers during development will likely influence morphogen diffusion. For instance, cell-rounding and loss of cell-cell adhesion during zebrafish morphogenesis could increase the extracellular cavity fraction to allow for faster diffusion[49,81]. Conversely, during later zebrafish somitogenesis, fluid-to-solid jamming transitions in the tissue architecture might severely inhibit fast extracellular diffusion[82]. Depending on the spatiotemporal scale of patterning, different tissues require different gradient ranges, and strongly hindered diffusion in densely packed epithelia without large extracellular cavities may allow for the slow day-long patterning timescales observed in

tissues such as the *Drosophila* wing disc[2]. Additionally, changes in binding partner numbers could alter the bound morphogen fraction to further influence morphogen diffusion.

We observed that binding differed between the Nodals Cyclops and Squint. First, the bound fractions were higher for Cyclops than for Squint, and binding times of Cyclops were on average 5 s longer. This contrasts with measurements showing that Squint binds to the Type II receptor Acvr2b-a with higher affinity than Cyclops[14]. Second, we observed that bound Squint molecules frequently exhibited slow diffusion along the membrane, while bound Cyclops molecules were mostly localized within a small area and effectively immobilized. Our single-molecule measurements are oblivious to the molecular identity of the morphogen binding partners and therefore represent a neutral description of overall cell surface binding. The differing mobilities of bound Cyclops and Squint might reflect differences in how both members of Nodal bind to components of the extracellular matrix. There are numerous potential extracellular binding partners, for example the EGF-CFC co-receptor Oep or other immobilized diffusion regulators such as heparan sulfate proteoglycans[7,8,17–21,69,83]. The different degrees of hindered diffusion that we observed – shorter residence times and lower fraction of bound Squint compared to Cyclops – are likely to underlie the different ranges of Nodal gradients (short-to-mid range for Squint and ultra-short range for Cyclops)[25,84–87].

Interestingly, while we also observed labeled Nodals and Leftys in the cytoplasm (excluded in the present analysis) in addition to their extracellular localization, we only rarely (in approximately 3 out of 100 movies) observed events where single molecules clearly passed the membrane and entered the cytoplasm. This indicates that internalization of Nodals/Leftys is a rare event, as opposed to the prolonged binding of Nodals on the cell surface, which provides a possible explanation for the hour-long half-lives of Nodals/Leftys in living zebrafish embryos[25,41,88]. Rare internalization also contrasts with the transcytosis mechanism described for the TGF-β superfamily ligand Dpp in *Drosophila*, in which repeated rounds of exocytosis and endocytosis lead to morphogen dispersal[72]. The differences in the dispersal mechanisms might be explained by the different time scales required for patterning of the zebrafish embryo (hours) and the *Drosophila* wing disc (days)[2].

In summary, we propose that Nodal and Lefty spreading follows a compartmentalized hindered diffusion model, in which cell-cell interfaces provide a confined, obstructive environment with restricted diffusion in particular for Nodal, whereas long-distance spreading of morphogens occurs within cavities between cells.

## Methods

### Zebrafish husbandry

The research was performed in accordance with all relevant ethical regulations. Wild Indian Karyotype (WIK) and TE zebrafish were maintained according to the guidelines of the EU directive 2010/63/EU, the German Animal Welfare Act and the State of Baden-Württemberg (Germany) and approved by the Regierungspräsidium Tübingen and the Regierungspräsidium Freiburg. Zebrafish were maintained exclusively for breeding and experiments were performed exclusively on zebrafish embryos.

### Generation of constructs

The designs for the HaloTag-tagged zebrafish Nodals Squint and Cyclops, Lefty1 and Lefty2 as well as secreted HaloTag were based on previously published GFP fusion constructs[25].

To generate pCS2-2xHA-HALO-3xGS, HaloTag was isolated using primers containing a tandem HA-Tag and primers containing a triple GS-linker, which was then cloned into the BamHI and XbaI sites of the pCS2 backbone[42].

HaloTag was then amplified from pCS2-2xHA-HALO-3xGS, and fusion constructs for Nodals and Leftys were generated using splicing-by-overlap-extension PCR[89–91] using the pCS2 backbone. The following primers were used:

For pCS2-2xHA-HALO-3xGs: GATCGGATCCATGTACCCATACGAT GTTCCAGATTACGCTGGATATCCATATGATGTTCCAGATTATGCTCGA GGAGCAGAAATCGGTACTGGCTT, GATCATCTAGAGATCGAGGCGCG CCGATCGATTAATTAAGCTTCCGGAGCCAGAACCTGAGCCGGAAATC TCGAGCG

For pCS2-Squint-HaloTag: GCAGGATCCCATCGATGCCACCATG TTTTCCTGCGGGCTCC, GGCTCGAGAGGCCTTGAATTCTCAGTGGCA GCCGCATTCTGC, CTCGAGATTTCCGGCGGATCCGCAGCAGCAG, CT GCTGCTGCGGATCCGCCGGAAATCTCGAG, GATCCACCGGTACCACC GGAGCAGAAATCGGTAC, GTACCGATTTCTGCTCCGGTGGTACCGGT GGATC

For pCS2-Cyclops-HaloTag: GCAGGATCCCATCGATGCCACCATG CACGCGCTCGGAGTCGC, GGCTCGAGAGGCCTTGAATTCTCACAGGC ATCCGCACTCCTC, GCCGCCGGGGGCCAGGAGCAGAAATCGGTAC, G TACCGATTTCTGCTCCTGGCCCCCGGCGGC, CTCGAGATTTCCGGCC CTGTCAGGAGCCCAG, CTGGGCTCCTGACAGGGCCGGAAATCTCGAG

For pCS2-Lefty1-HaloTag: GCAGGATCCCATCGATGCCACCAT GACTTCAGTCCGCGCCG, CTATAGTTCTAGAGGCTCGAGTCAGCCGG AAATCTCGAG, GATCCACCGGTCGCCACCGGAGCAGAAATCGGTAC, GTACCGATTTCTGCTCCGGTGGCGACCGGTGGATC

For pCS2-Lefty2-HaloTag: GCAGGATCCCATCGATGCCACCAT GGCTCTGTTCATCCAGC, CTATAGTTCTAGAGGCTCGAGTCAGCCGG AAATCTCGAG, CCCTCCAGTCCTGGGCGGAGCAGAAATCGGTAC, GTA CCGATTTCTGCTCCGCCCAGGACTGGAGGG

To generate pCS2-secreted-HaloTag, HaloTag was amplified with the primers GATCCACCGGTACCACCGGAGCAGAAATCGGTAC and CTATAGTTCTAGAGGCTCGAGTCAGCCGGAAATCTCGAG and cloned by restriction digest using AgeI and XbaI.

pCS2-secreted-HaloTag-GFP was generated based on pCS2-secreted-HaloTag. GFP was fused to the C-terminus of the HaloTag (separated by a GS linker) via splicing-by-overlap-extension PCR with the following primers: CCACCGGTACCACCGGAGC, GCCTTGAATT CTCACTTGTACAGCTCGTCC, CTCGCCCTTGCTCACGGATCCGCCGG AAATC, GATTTCCGGCGGATCCGTGAGCAAGGGCGAG. The fused construct was cloned by restriction digest using AgeI and XbaI afterwards.

### mRNA synthesis

For capped mRNA synthesis, pCS2-Cyclops-HaloTag, pCS2-Squint-HaloTag, pCS2-Lefty1-HaloTag, pCS2-Lefty2-HaloTag, pCS2-secreted-HaloTag, pCS2-Squint-GFP[25] and pCS2-mem-GFP[44] were linearized with NotI, and a mMESSAGE mMACHINE SP6 Kit (Invitrogen) was used for in vitro transcription according to the manufacturer's recommendations. To generate mRNA encoding Oep, pCDNA3-oep-FLAG was linearized with NotI and transcribed using an mMESSAGE mMACHINE T7 Kit (Invitrogen) according to the manufacturer's recommendations.

### Quantitative reverse transcription PCR (qRT-PCR)

To validate the biological activity of the Halo-tagged Nodals and Leftys, we performed qRT-PCR assays for the Nodal target gene *goosecoid* (*gsc*). The previously characterized GFP fusion constructs[25] were used as positive controls, and the mRNAs for each morphogen and their corresponding fusions were synthesized in parallel. The qRT-PCR was performed as described in[25] with the following changes: The total RNA was extracted using NucleoZOL (Macherey-Nagel). Up to 1 μg of purified total RNA was used to synthesize cDNA using the SuperScript III First-Strand Synthesis SuperMix (Thermo Fisher). The Platinum SYBR Green qPCR SuperMix-UDG (Thermo Fisher) was used in a Light-Cycler® 96 Instrument (Roche) for 45 cycles using 2-step amplification. Quantification cycle (Cq) values were obtained using the LightCycler® 96 software.

### Confocal fluorescence microscopy

TE embryos were injected at the one-cell stage with 50 pg of mRNA encoding secreted-HaloTag, Squint-HaloTag or Cyclops-HaloTag or 60 pg of mRNA encoding Lefty1-HaloTag or Lefty2-HaloTag. Embryos were proteolytically dechorionated using 10 mg Pronase (Sigma Aldrich) in 10 ml Danieau's medium and washed with Danieau's medium to remove the Pronase. The embryos were then incubated in a 1:5000 dilution of TMR HaloTag Ligand (5 mM; Promega) in Danieau's medium. After 30–60 min at 28 °C, they were rinsed with embryo medium twice and mounted in a glass bottom dish using 1.5% low-melting agarose. Imaging was performed with an LSM 780 NLO (ZEISS) system using an LD LCI Plan-Apochromat 25×/0.8 Imm Korr DIC objective to acquire animal views at a depth of approximately 35 μm into the tissue.

### Sample preparation for single-molecule imaging

Embryos were dechorionated directly after fertilization using 10 mg Pronase (Sigma Aldrich) in 10 ml Danieau's medium and washed with Danieau's medium to remove the Pronase. To express morphogen constructs, embryos were injected at the 1-cell stage with 1 pg of mRNA encoding Squint-HaloTag, Lefty1-HaloTag, Lefty2-HaloTag, secreted HaloTag, or 2 pg of mRNA encoding Cyclops-HaloTag together with

10 pg of mRNA encoding memGFP into the animal pole. The diameter of the injection mix droplet was measured with a stereo microscope (Olympus SZX2-ZB10) equipped with a camera (CAM-SC50) and the CellSens Entry 2.3 (Build 18987) imaging software (Olympus), and set to 124 μm, corresponding to a droplet volume of 1 nl, by adjusting the injection duration. For Oep experiments, 1 pg of mRNA encoding HaloTag-Squint was injected together with 0.3 pg, 3 pg, or 30 pg of mRNA encoding Oep and 10 pg of mRNA encoding memGFP.

To label the HaloTag, embryos at the 2-cell stage were placed into two separate glass tubes, and most of the embryo buffer was removed such that embryos were just covered sufficiently. 1 ml of 5 nM HaloTag-JF549[45] dye solution was then added into one of the tubes and 10 nM HaloTag-JF549 dye solution into the other tube and incubated for 30 min. After staining, zebrafish were washed with Danieau's buffer, followed by two additional washing steps each after 15 min.

Embryos were incubated at room temperature (22 °C). We monitored the transition from the 64-cell to the 128-cell stage and visually identified synchronously developing embryos. 6–8 embryos at the 128-cell stage stained with 10 nM dye solution were then mounted onto the microscope by placing them into a glass bottom dish with a thickness of 0.17 mm (Delta T, Bioptechs, Butler, PA). If the density of visible fluorescent molecules was too high, the mounted embryos were exchanged with embryos stained with 5 nM dye solution. On the microscope, embryos grew further at room temperature. Transitions between embryo stages were counted by identifying cytoplasmic divisions (cytokinesis events), when the GFP-stained cell membrane grows inward until cell division. Fluorescence imaging was stopped when embryos reached the sphere stage.

## Reflected light-sheet microscopy (RLSM)
Single-molecule imaging of live zebrafish embryos was carried out using a custom-built reflected light-sheet microscope[63] with modifications to image live zebrafish embryos[42]. The microscope was built around a commercial Nikon TI microscope body equipped with a water-immersion objective (60× 1.20 NA Plan Apo VC W, NIKON), a dichroic mirror (F73–866/F58–533, AHF), an emission filter (F72–866/F57–532, AHF), a notch filter (F40–072/F40–513, AHF) and an EM-CCD Camera (iXon Ultra DU 897U, Andor). Fluorescence light was post-magnified by a factor of 1.5× before reaching the camera chip, which resulted in a pixel size of 166 nm. The microscope was controlled using the NIS Elements software Version 4.40.00 64 bit (Nikon) and a NIDAQ data acquisition card (National Instruments).

Reflected light-sheet illumination was achieved using a custom-built tower mounted above the sample dish. AOTF (AOTFnC-400.650-TN, AA Optoelectronics) controlled laser light of 488 nm (IBEAM-SMART-488-S-HP, 200 mW, Toptica) and 561 nm (Jive 300 mW, Cobolt) was coupled into the tower via a single-mode fiber, where it was focused by a cylindrical lens into the back-focal plane of a water-dipping objective (40 × 0.8 NA HCX Apo L W, Leica) and subsequently reflected by the chip of an AFM cantilever (custom-coated cantilever based on model: HYDRA2R-100N-TL-20 but with both sides coated in 40 nm Al). The resulting light-sheet had a thickness of approximately 3 μm. The laser power of the light-sheet was 40 mW for the 561 nm laser and 4 mW for the 488 nm laser.

Compared to previous measurements with the mEos2 label[42], HaloTag together with JF549 allowed for faster frame cycle times and longer tracks. Movies of morphogens were recorded with 10 ms exposure time and a total frame cycle time of 11.7 ms. The illumination time was set to match the exposure time.

For continuous movies, a sequence starting with 10 frames with 488 nm laser illumination showing the membrane-GFP signal was recorded, followed by 1000 frames of 561 nm laser illumination to image single morphogen molecules and finalized by another 10 frames of 488 nm illumination.

For time-lapse microscopy movies of Squint-HaloTag and Cyclops-HaloTag, frames were illuminated for one frame with the 561 nm laser followed by one frame with 488 nm laser illumination. Dark times of different lengths were introduced between illuminated frames, resulting in frame acquisitions every 58 ms, 199 ms or 1006 ms (time-lapses) for which a total number of 200, 59 and 24 illuminated frames were recorded, respectively.

## Splitting movies into reference channel and single-molecule channel
Movies containing frames with memGFP signal as well as frames with signal of Halo-tagged morphogens were separated using TrackIts movie splitter. Two separate movies were obtained, one containing only the memGFP signal and one containing only single-molecule signal. Movies were discarded if considerable drift due to embryo movement was evident in order to guarantee a correct classification of tracks to their region classes (cavity, interface).

## Segmentation of extracellular regions using a CNN
A U-Net[92] CNN was trained to segment extracellular regions of developing zebrafish embryos based on the memGFP intensity using ZeroCostDL4Mic[48], a state-of-the-art image segmentation platform. Target images for training were created by first using TrackIt's sub-region drawing tool to manually draw outlines around the extracellular space of 452 images. The "Average frames" function was used to average all frames of a memGFP movie, and the "Gaussian filter" function with a kernel size of 1 px was applied to smoothen the averaged image. A custom Matlab script was then used to create a U-Net compatible 8-bit.tif file containing the training masks.

Training was performed in the cloud using the Google Colaboratory platform provided by DL4Mic. The model was trained over 200 epochs on 90% of the training images while 10% of images were used for validation. This resulted in a final training loss of 0.163. The trained model was downloaded and integrated into a custom Python program to create regions of interest compatible with our single-molecule tracking software TrackIt[47]. All frames of the memGFP movies in a folder were averaged and padded with zeros to match the U-Net network requirements. The images were then segmented with the trained model, and binary masks were created by applying a user-defined intensity threshold between 0-255, which was set to 240. Polygonal regions of interest with a minimum size of 150 px were then saved in a TrackIt compatible .roi file.

Once loaded into TrackIt, the segmentation results were visually quality-controlled for each movie. Parts where the memGFP signal was blurry (e.g. when lying far away from the edge of the zebrafish), or parts where the laser light was blocked or absorbed, were either adjusted or cut-off manually.

A second region containing extracellular cavities was added by manually selecting parts of the regions that had been segmented by the CNN (see for example Fig. 1d). Cavities were defined as areas, where the cell membranes of more than two cells meet and the memGFP signal of all cells are distinguishable. Interfaces were defined as areas, where two cells are aligned in a way that the memGFP signal of the cell membranes of both cells overlap and are not distinguishable.

## Tracking of single-molecule microscopy data for mobility analysis of morphogens
Single-molecule microscopy data of Halo-tagged Cyclops, Squint, Lefty1 & Lefty2 and secreted-HaloTag were analyzed with TrackIt[47]. A threshold factor of 1.5 was used to detect single-molecule events. The nearest neighbor algorithm was used with a tracking radius of 10 px (= 1.66 μm) to link single-molecule detections into tracks, and 1 gap frame was allowed to bridge detection gaps if a molecule was already detected for at least 2 consecutive frames. TrackIt's "Delete tracks touching borders" option was used, which means that tracks were

assigned to regions if they lied completely inside the region of interest while tracks crossing region borders were discarded and treated as non-linked detections.

## Distribution of jump distances and diffusion analysis

For diffusion analysis, TrackIt's data analysis tool was used to fit the cumulative distribution of jump distances with a three-component Brownian diffusion model. The total number of bins of the cumulative jump distance histogram was set to 1660 corresponding to a bin size of 1 nm. Generally, two-dimensional tracks acquired in single-molecule tracking are a projection of three-dimensional motion, which underestimates diffusion coefficients. Biases also arise from molecules moving out of the depth of focus of the objective (approx. 0.7 μm), which overestimates the fraction of bound molecules[93,94]. We did not correct for these effects in our analysis. However, since all morphogens/conditions will experience similar bias, the comparisons between species we discuss will not be affected. To prevent an overrepresentation of bound molecules, a maximum of 10 jumps was considered per track. Jumps over gap frames were not considered. The errors of diffusion coefficients $D_{1,2,3}$ and fractions $A_{1,2,3}$ were estimated by repeating the analysis 500 times using random samples of 50% of the jump distances, and the standard deviation of the resulting diffusion coefficients and fractions were calculated. To assess whether a two- or three-component model best describes our data, we compared the reduced chi-squared of the 2-rate and 3-rate model fits and furthermore used the Akaike Information Criterion (AIC)[58]. For model comparison, we calculated the difference in AIC values using the residual sum of squares (RSS) obtained from the least squares fit[95]:

$$\Delta AIC_i = \left(2k_{2rate} + n\ln(RSS_{2rate})\right) - \left(2k_{3rate} + n\ln(RSS_{3rate})\right) \quad (1)$$

where $k$ is the number of parameters used in each model and $n$ is the number of data points.

To visualize the diffusion analysis results, the probability distribution $p(r)$, as obtained from the fit results, was plotted together with the histogram of jump distances using

$$p(r) = \frac{1}{2\tau} r \cdot \Delta r \left( \frac{A_1}{D_1}\exp\left(\frac{-r^2}{4\tau D_1}\right) + \frac{A_2}{D_2}\exp\left(\frac{-r^2}{4\tau D_2}\right) + \frac{\frac{A_3}{D_3}\exp\left(\frac{-r^2}{4\tau D_3}\right)}{1 - \exp\left(\frac{-r_{tr}^2}{4\tau D_3}\right)} \right)$$

$$(2)$$

where $r$ is the jump distance, $\Delta r$ is the bin width of the jump-distance histogram (here 20 nm), $\tau$ is the frame cycle time and $D_i$ and $A_i$ are the diffusion coefficients and fractions resulting from the cumulative jump distance distribution fit. The last term is normalized by $\left(1 - \exp\left(\frac{-r_{tr}^2}{4\tau D_3}\right)\right)$, with $r_{tr}$ representing the tracking radius, to account for the cut-off due to the lower and upper limit of jump distances.

## Jump angle analysis

Jump angles $\theta$ were calculated from two consecutive jumps (three consecutive localizations)[47]. The angle $\theta$ indicates the change in direction of a molecule after it has moved in one direction (see inset in Fig. 3e). While an angle of 0° indicates no change, i.e., a direct forward movement of the molecule, an angle of 180° indicates that a molecule moved backward in the opposite direction. Angles <180° or >180° indicate a movement to the right or left, respectively. The degree of reverse motion was quantified by calculating the 'fold anisotropy' metric, $f_{180/0}$[60], which measures how many-fold more likely a backward jump is, compared to a jump forward, using:

$$f(180/0) = \frac{BWD}{FWD} = P\left(\frac{180° \pm 30°}{0° \pm 30°}\right) \quad (3)$$

where BWD is the probability for a backward jump with an angle $\theta$ of 150°–210° and FWD is the probability for a forward jump with an angle $\theta$ of 330°–30°.

To calculate the angle between a jump and the region border, we first identified the point of the polygonal ROI that was closest to both detections of a jump. We then calculated the average of the orientation of the two neighboring polygonal border segments. Subsequently we determined the acute angle between the track segment and the ROI border segment.

## Mean squared displacement (MSD) analysis

For MSD analysis, tracks with a minimum duration of 20 frames were considered. The first 10 points of the MSD curves were fitted with the linear function $MSD = 4 \cdot D \cdot \tau + c$, where $D$ is the diffusion coefficient and $c$ is a constant to account for the localization error.

## Spot density ratio

The density of single-molecule detections was calculated for each movie and region class separately by dividing the number of detections in each of the regions by its number of pixels. The ratio between the spot density in the cavity region and the spot density in the interface region was then calculated for each movie. Movies containing no detections in one of the regions or movies with only one region class were discarded. p-values were calculated using the non-parametric Kruskal-Wallis-Test using GraphPad Prism 9.0.1.

## Analysis of time-lapse microscopy data

Tracking settings were optimized for the nearest-neighbor algorithm to track only bound molecules based on the time spend within a certain area by using small tracking radii in combination with larger minimum track lengths. TrackIt's automatic tracking radii prediction tool was used to ensure equal tracking-loss probabilities due to tracking errors and photobleaching across all time-lapse conditions. The resulting tracking radii for a loss probability of 0.005 were: 1.55 pixels or 257 nm (continuous), 2.05 pixels or 340 nm (58 ms time-lapse), 2.92 pixels or 485 nm (200 ms time-lapse) and 3.62 pixels or 601 nm (1 s time-lapse). To ensure that no freely diffusing molecules were tracked and to minimize false connections, a minimum track length of 5 frames was used for continuous and 3 frames for time-lapse movies. Other tracking settings were as described above.

Fluorescence survival time distributions of Squint and Cyclops were extracted from the single-molecule tracks, and GRID[61] was used to determine the dissociation rate spectrum. In brief, GRID uses a superposition of 40 exponential functions with fixed decay rate constants between $10^{-2}$ s$^{-1}$ and $10^2$ s$^{-1}$ and an appropriate set of regularizations to ensure robust convergence of the fit to the survival time distributions. As a result, GRID yields the amplitudes corresponding to each of the fixed decay rates. Binding times were calculated as the inverse of the dissociation rate. The tails of the survival time distributions were cut off below a probability of 0.01 due to a low number of events.

The rate spectrum is a measure for how often dissociation events of a certain dissociation rate population occur within a specific time (Supplementary Fig. 6b). This can be converted into a "state" spectrum by dividing the fractions of the event spectrum with the respective dissociation rates. This results in the distribution of binding states at any given point in time. To estimate errors of the dissociation rate spectra, a 500× resampling was performed with randomly selected 80% of the data. Boundaries for dissociation rate clusters were then manually assigned to calculate the standard deviation of the results.

## Agent-based modelling

Agent-based models were implemented in Python3[96]. Morphogens were modeled as agents performing a random walk on a

two-dimensional simulation geometry. The geometry was generated in Fiji[97] by scaling the binary masks of extracellular space such that each pixel in the image was 10 nm × 10 nm. Reflective boundary conditions were used. All intracellular pixels were set to 0 and extracellular to 1. The extracellular space was populated with 100 morphogens at random starting positions with a fraction of morphogens in bound state. The initial bound fraction ($BF_I$) was estimated using an empirical function of the receptor density ($\sigma$), residence time ($\tau$), and binding strength ($S$) to ensure that the equilibrium bound fractions were achieved quickly during the simulation.

$$BF_I = \frac{\tau \cdot S}{50 \cdot \log_{10}\left(\frac{10}{\sigma}\right)} \qquad (4)$$

Morphogens were allowed to occupy only the extracellular positions. The receptors were placed at the boundary of the extracellular region with uniform spacing based on the required receptor density. The morphogen position and state (bound or unbound) was updated at each simulation step (10,000 steps of 10 ms each). To simulate random walks of morphogens, their jump distances ($r$) were drawn from a range of jumps with probabilities given by the probability density function:

$$p(r) = \frac{f(r)}{\sum_x f(x)} \qquad (5)$$

given that

$$f(r) = \frac{r}{2 \cdot D \cdot \tau} \cdot e^{\left(\frac{-r^2}{4 \cdot D \cdot \tau}\right)} \qquad (6)$$

where $D$ is the diffusion coefficient (either $D_{free}$ or $D_{bound}$), $\tau$ is the timestep (10 ms) and $r, x \in [0,2]$.

The free and bound diffusion coefficients were set to 30 µm²/s ($D_{free}$) and 0.5 µm²/s ($D_{bound}$) as estimated from the experimental observations. A new position in the extracellular space was picked randomly from the available positions at distance $r$ from the current position. The morphogen state was changed from unbound to bound if the distance to a receptor was ≤ 20 nm. The morphogen stayed bound for the time $t_b = \tau \cdot \log_{10}\left(\frac{1}{\upsilon}\right)$, where $\upsilon$ is a random number between 0 and 1, and $\tau$ is the residence time. The jump and angular histograms and localization plots were generated in a similar manner as the experimental dataset to validate the model.

For the parameter screen, the two-dimensional simulation geometry was modified to narrow or widen the extracellular space. This was achieved by iteratively changing the pixel values at the extracellular boundary. For each iteration, the width of the extracellular space was changed by 20 nm, thereby changing the extracellular fraction ($\eta = \frac{\text{Extracellular Area} \times 100}{\text{Total Area}}$). Five $\eta$ values ranging from 8% to 16%, three receptor densities $\sigma$ (0.05, 0.5, 5.0 µm⁻¹) and 5 residence times $\tau$ (0, 1, 6, 11, 16 s) were tested.

**Morphotrap experiments and image analysis**
Images of zebrafish embryos injected with different amounts of membrane-tethered GFP-binding nanobody were acquired as described previously[69]. Image analysis was performed manually in Fiji 2.9.0[97]. The GFP channel of the confocal images was converted into a.tif file. Five regions of interest (ROIs, 64 × 171 pixel) were manually selected from the image. For each ROI, 15 cavity and interface regions were selected, and the mean gray value was measured to quantify the GFP localization in each region. The ratio of the mean gray value of the cavity to that of the interface was used as $\rho_c / \rho_i$.

**Reporting summary**
Further information on research design is available in the Nature Research Reporting Summary linked to this article.

## Data availability
All single-particle tracking data and simulated tracks are freely available at Dryad [https://doi.org/10.5061/dryad.9kd51c5kg]. Source data for figures are provided with this paper in 'Source Data.xlsx'. Data supporting the findings of this manuscript will be available from the corresponding authors after publication upon reasonable request.

## Code availability
The TrackIt software is freely available. TrackIt was written in Matlab and is available on GitLab [https://gitlab.com/GebhardtLab/TrackIt] and Zenodo [https://doi.org/10.5281/zenodo.7092296]. The code for the agent-based model is available on GitHub [https://github.com/mueller-lab/morphogenDiffusion-ABM] and Zenodo [https://doi.org/10.5281/zenodo.7104354].

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

## Acknowledgements

We thank Gilbert Weidinger (Ulm University) for his constant support by providing access to his zebrafish facility, members of the Gebhardt and Michaelis labs for helpful discussions, Karlotta Bosch and Maximilian Haas for help cloning the HaloTag constructs, Astrid Bellan-Koch for help cloning the secHalo-GFP, and Catrin Weiler and Marianne Wiechers for generating mRNAs. The work was funded by the European Research Council (ERC) under the European Union's Horizon 2020 Research and Innovation Program (No. 637987 ChromArch to J.C.M.G., No. 637840 QUANTPATTERN to P.M., No. 863952 ACE-OF-SPACE to P.M.) and the German Research Foundation (No. 422780363 SPP 2202 GE 2631/2–1 and No. 427512076 GE 2631/3–1 to J.C.M.G.). Support by the Collaborative Research Centre 1279, the Center for Translational Imaging MoMAN of Ulm University, the DFG Centre for the Advanced Study of Collective Behaviour (EXC 2117) and the International Max Planck Research School "From Molecules to Organisms" is acknowledged.

## Author contributions

P.M. and J.C.M.G. conceived the project; T.K., D.M., P.M., and J.C.M.G. designed the project; D.M. cloned morphogen fusion proteins and performed confocal microscopy; D.C. and A.N.L. validated the activity of the fusion proteins with injections and qRT-PCR; T.K. performed the single-molecule measurements; J.C. contributed to the single-molecule measurements; T.K. programmed TrackIt; J.G. and T.K. created the CNN segmentation tool; T.K. analyzed data with contributions from J.C. and J.C.M.G.; A.N.L. performed simulations; T.K., A.N.L., P.M., and J.C.M.G wrote the manuscript with comments from all authors.

## Funding

## Competing interests

The authors declare no competing interests.
