## [Peer Review File · Nature Communications]

Single-molecule tracking of Nodal and Lefty in live zebrafish embryos supports hindered diffusion modelEditorial Note: Parts of this Peer Review File have been redacted as indicated to remove third-party material where no permission to publish could be obtained.

REVIEWER COMMENTS

Reviewer #1 (Remarks to the Author):

Summary -

Kuhn et al. show that zebrafish Nodals (Cyclops and Squint) and Leftys (Lefty1 and Lefty2) exhibit extracellular movement congruent with the hindered diffusion model. By injecting mRNA encoding Halo Tag fusions for these proteins, the authors image and track these proteins at single molecule level. They find that there are differences in movement in intercellular spaces and in "cavities" such as where three cells meet, where Nodals and Leftys move more freely due to the lack of binding and unbinding to the receptors present at cell interfaces. The authors then test their expectation that increasing binders would affect Nodal and Lefty localization and/or diffusion by injecting different amounts of mRNA for the co-receptor oep. Analysis of their data agrees with hindered diffusion model where diffusion coefficients for Nodals and Leftys do not change but the fraction of Nodals and Leftys found in interfaces and that in cavities change depending on binding/ unbinding to receptors. Furthermore, their modeling reveals that the geometry of the tissue, width of the intercellular spaces, in addition to receptor density and residency time also affects morphogen movement in extracellular space.

Significance -

The mechanisms driving morphogen mediated patterning is one of the major questions in development. How morphogens move in extracellular spaces to send signals to cells spanning long distances is key to understanding an aspect of this process and multiple models have been proposed in the literature. This manuscript provides data supporting the "simplest" of these models- hindered diffusion. While the conclusions are not conceptually new, this paper presents first-of-its-kind quantitative data on zebrafish Nodals and Leftys at the single molecule level, which to date has not yet been shown in vertebrate embryos for any morphogen to my knowledge. With careful measurements through single molecule imaging and tracking, the authors were able to show dynamics of morphogen behavior with millisecond resolution that support a model in which morphogen diffusion can be explained based on geometrically-constrained diffusion interspersed with binding-unbinding events. Establishing using Halo Tag fusion proteins and reflected lightsheet microscopy in zebrafish embryos is a useful approach that could be used for other morphogens. It is nice to see such direct data in support of a conceptually plausible model so I found this paper exciting.

Comments:

Discussion -

The authors frame their findings as fitting the hindered diffusion model, where geometry and binding impact morphogen movement. Their "counterintuitive" finding of accumulation in cavities was explained with the model. Discussion of the implications for this cavity accumulation for different tissues (eg. densely packed or growing etc.) and in developmental patterning context would be useful. And discuss, all in all, whether there are alternative interpretations. As written, the paper comes across as a "confirmation" of a model but without further implications which I think undersells the nice data.

Data presentation -

All the data regarding the secreted Halo Tag control are presented in the supplement. It would be good to show as control the secreted Halo Tag data alongside the main figures and corresponding measurements for Halo Tag- Sqt/Cyc/Lefty1/Lefty2 as there is always a concern as to how the tag affects the properties of the tagged protein.

Clarifications -

How were binding/unbinding events distinguished from movement through tortuous interface without binding and unbinding? Please clarify in text.

Calculating an angle from the jump distance is explained in text as "calculated the angles within a

track spanned by three consecutive localizations⁵¹.” (lines 170-171) Providing a clarification sentence about how the measurement was done would be useful - was there a reference segment from which the angle was measured? How was it distinguished to measure either an acute or obtuse angle as an angle between three points could give you both?

GRID as the acronym is first mentioned in line 201. The authors should briefly describe the method used for measuring survival time distributions.

Why is the three component model (as opposed to two or one) is the most reasonable fit could be explained better. Obviously more parameters gives a better fit but what is the justification?

What were the criteria for manual identification of the cavities? Was there a width/size threshold? The top cavity in the first panel in figure 2a has a cavity region that is different in shape from the other cavities shown.

Figure 6 d (ii), on the leftmost panel (0pg) there is still quite a bit of signal (possibly autofluorescence?). Please address.

Concerns - please discuss the limitations of the study regarding the following -

Validation of the Halo Tag attached versions of these proteins, whether they would reflect endogenous behavior and function of these morphogens is incomplete. In supplemental figure 1, authors show that the defects from injection with Sqt-HaloTag are similar to those from injection with Sqt-GFP. Please show any assays done to validate Cyc-HaloTag, Lefty1-HaloTag, and Lefty2-HaloTag. A better control would be to show that the range of action of the fusions is equivalent to the untagged counterparts since this would control for morphogen dispersal rather than just activity. Measuring expression of downstream genes (such as *gsc* or other nodal responsive genes) following mosaic expression of HaloTag tagged versions and untagged mRNA could be an additional approach.

Some of the interpretations of the data such as finding more morphogen-HaloTag in cavities than in interfaces and rarely observing single molecules entering the cytoplasm (line 343-344) rely on the aspects of the HaloTag tagged version of the protein not validated here, such as how or whether they get cleared, endocytosed, degraded etc. HaloTag fused to something secreted that is similar in size to Nodal and Lefty (e.g. GFP) would be an appropriate control for comparing the diffusion behavior.

Citations -

Should cite the following paper which shows a similar approach: Greiss, F., Deligiannaki, M., Jung, C., Gaul, U., & Braun, D. (2016). Single-Molecule Imaging in Living *Drosophila* Embryos with Reflected Light-Sheet Microscopy. *Biophysical journal*, 110(4), 939-946. <https://doi.org/10.1016/j.bpj.2015.12.035>

Reviewer #2 (Remarks to the Author):

In this excellent study, the authors apply single-molecule imaging and tracking in an in vivo context to directly study how the geometry and binding landscape in the extracellular environment of a developing tissue affects the effective diffusivities of different morphogens. They quantitatively characterize diffusive microstates of single morphogen molecules and disentangle the roles of surface receptor binding and geometry through well-conceived follow-up experiments and a simple but helpful computational model. Taken together, their work supports the hindered diffusion model of differential diffusivity, which marks an important contribution to ongoing efforts in developmental biology aimed at understanding the mechanisms of both Positional Information and Turing-style self-organized pattern formation.

The experimental results are well-presented and carefully interpreted, and the manuscript overall is clearly and concisely written and highly polished. Its publication will help popularize single-molecule imaging in cell and developmental biology, where such techniques are currently widely considered

inaccessible. It will thus be of interest to a broad range of biologists for both its methodological advances and biological findings.

Given the above, I fully support the article's speedy publication.

Comments on content

- The inferred diffusion coefficients should be compared to those derived from traditional methods, such as FCS. The authors argue that the results are consistent with the literature insofar that the different morphogens are very similar to each other in terms of their free diffusivity, but the actual magnitudes of the coefficients are not being compared. More specifically, it would be relevant to state whether the coefficients of the unbound fractions (D3) match what has been reported from FCS measurements on freely diffusing molecules.
- The light sheet is approximately 3 microns thick, which raises the question whether the analysis needs to account for molecular motion in the z-axis. This may not be necessary or may be addressed within the TrackIt software, but either way the authors should clarify this point.
- Regarding the diffusion model fitting in Fig 3 and Supp Fig 4: The fact that the SSEs are lower in the 3-component model may be trivial given the model's additional degrees of freedom compared to the 2-component model. It would be more convincing if a measure was used that corrects for the number of parameters being estimated, such as the Akaike Information Criterion (AIC). Furthermore, the text does not make entirely clear what the authors' interpretation is for each of the three components (D1=bound, D2=?, D3=free).
- Regarding the angles (Fig 3e), it would be interesting to show how they relate to the angle of the relevant cell-cell interface (or the nearest boundary for cavities). Presumably, constrained diffusion would lead to more frequent reverse motion for angles more orthogonal to the boundary than for angles more parallel to the boundary. This would strengthen the authors' interpretation of the prevalence of reverse motion.
- In the methods, please state the thickness of the glass-bottom dishes used, as this is important for the optics.

Comments on presentation

- Fig 4c and Supp Fig 6b: It would be easier to read these figures if the y-axes were labeled with percentages (1%, 10%, 100%).
- Fig 6 and Supp Fig 9: The colormaps used are not perceptually uniform, which is no longer considered Good Practice in data visualization and should be changed. For the localization density plots in particular, the reader's perception of the patterns would be more accurate with a perceptually uniform colormap.
- Fig 6b: This 3d figure is not easy to read, but I acknowledge that there is no real solution for this problem. Consider showing three 2d plots instead, one for each level of sigma (same ordering as 6c).

Details

- Line 157: Use of the term "significant" should be avoided absent a statistical test for significance. Note that the resampling approach used to generate the s.d. in Fig 3 (and elsewhere) can also be

used to perform hypothesis testing (see e.g. the tidymodels package "infer" for R), which would be applicable here. As the results are very clear, rewording would also be sufficient in this case.

- Line 209: "remarkable agreement" seems to be an overstatement, given the differences in inferred and predicted Squint binding times (11s vs 4s).

- Typo in Supp Fig 8: "a)" is repeated

- Typo in Line 328: "where" should be "were"

Response to reviewers' comments

We thank the reviewers for their helpful and constructive criticism. We highlighted the corresponding changes to text passages in the manuscript in yellow. Below, we address each point raised by the reviewers individually.

Reviewer #1:

Summary –

Kuhn et al. show that zebrafish Nodals (Cyclops and Squint) and Leftys (Lefty1 and Lefty2) exhibit extracellular movement congruent with the hindered diffusion model. By injecting mRNA encoding Halo Tag fusions for these proteins, the authors image and track these proteins at single molecule level. They find that there are differences in movement in intercellular spaces and in “cavities” such as where three cells meet, where Nodals and Leftys move more freely due to the lack of binding and unbinding to the receptors present at cell interfaces. The authors then test their expectation that increasing binders would affect Nodal and Lefty localization and/or diffusion by injecting different amounts of mRNA for the co-receptor oep. Analysis of their data agrees with hindered diffusion model where diffusion coefficients for Nodals and Leftys do not change but the fraction of Nodals and Leftys found in interfaces and that in cavities change depending on binding/ unbinding to receptors. Furthermore, their modeling reveals that the geometry of the tissue, width of the intercellular spaces, in addition to receptor density and residency time also affects morphogen movement in extracellular space.

Significance –

The mechanisms driving morphogen mediated patterning is one of the major questions in development. How morphogens move in extracellular spaces to send signals to cells spanning long distances is key to understanding an aspect of this process and multiple models have been proposed in the literature. This manuscript provides data supporting the “simplest” of these models- hindered diffusion. While the conclusions are not conceptually new, this paper presents first-of-its-kind quantitative data on zebrafish Nodals and Leftys at the single molecule level, which to date has not yet been shown in vertebrate embryos for any morphogen to my knowledge. With careful measurements through single molecule imaging and tracking, the authors were able to show dynamics of morphogen behavior with millisecond resolution that support a model in which morphogen diffusion can be explained based on geometrically-constrained diffusion interspersed with binding-unbinding events. Establishing using Halo Tag fusion proteins and reflected lightsheet microscopy in zebrafish embryos is a useful approach that could be used for other morphogens. It is nice to see such direct data in support of a conceptually plausible model so I found this paper exciting.

We thank the reviewer for the positive assessment of our manuscript.

Comments:

Discussion –

The authors frame their findings as fitting the hindered diffusion model, where geometry and binding impact morphogen movement. Their “counterintuitive” finding of accumulation in cavities was explained with the model. Discussion of the implications for this cavity accumulation for different tissues (eg. densely packed or growing etc.) and in developmental patterning context would be useful. And discuss, all in all, whether there are alternative interpretations. As written, the paper comes across as a “confirmation” of a model but without further implications which I think undersells the nice data.

As suggested, we now also discuss the implications of cavity accumulation in dynamically changing tissue geometry during embryogenesis. We added the following paragraph (lines 344-355):

*“Cavity accumulation in the loosely packed zebrafish blastoderm may allow for the rapid morphogen transport required to accomplish tissue patterning within the short, hour-long timescales during early embryogenesis. Dynamic changes in tissue geometry and cell numbers during development will likely influence morphogen diffusion. For instance, cell-rounding and loss of cell-cell adhesion during zebrafish morphogenesis could increase the extracellular cavity fraction to allow for faster diffusion^{49,81}. Conversely, during later zebrafish somitogenesis, fluid-to-solid jamming transitions in the tissue architecture might severely inhibit fast extracellular diffusion⁸². Depending on the spatiotemporal scale of patterning, different tissues require different gradient ranges, and strongly hindered diffusion in densely packed epithelia without large extracellular cavities may allow for the slow day-long patterning timescales observed in tissues such as the *Drosophila* wing disc². Additionally, changes in binding partner numbers could alter the bound morphogen fraction to further influence morphogen diffusion.”*

Data presentation –

All the data regarding the secreted Halo Tag control are presented in the supplement. It would be good to show as control the secreted Halo Tag data alongside the main figures and corresponding measurements for Halo Tag- Sqt/Cyc/Lefty1/Lefty2 as there is always a concern as to how the tag affects the properties of the tagged protein.

We agree that the secreted Halo Tag data is an important control for our experiments, and we are now showing the data concerning the secreted Halo Tag in every panel of Fig. 3 and in Fig. 4c. Please note that we refrained from time-lapse measurements of the secreted Halo Tag alone, as the binding events of the secreted Halo Tag were much shorter than those of Cyclops and Squint (as can be seen in the survival time distribution in Fig. 4c and Supplementary Fig. 6a).

Clarifications –

How were binding/unbinding events distinguished from movement through tortuous interface without binding and unbinding? Please clarify in text.

The reviewer points to a caveat of our measurements. While we can classify bound molecules by their time they spend within a certain area through the use of restrictive tracking settings (small tracking radius in combination with a large minimum track length), we cannot identify the origin of binding (specific binding to a receptor or unspecific sticking within a tortuous environment) of individual tracks. Therefore, we performed the titration experiment of *oep* overexpression, which allowed us to conclude that *Oep* can directly hinder the diffusion of *Nodal*. We now emphasize the uncertainty of the origin of binding in the text (line 239-241 and line 251):

“While our single-molecule imaging approach enables us to classify bound molecules by their time they spend within a certain area, the origin of binding of individual tracks remains obscure. Binding might reflect both specific interactions with receptors or unspecific retention by the tortuous environment.”

and

*“Thus, our data show on a single-molecule level that Squint at least to some extent binds to *Oep*, which can directly hinder the diffusion of *Nodal* by transiently trapping the morphogen on the membrane.”*

Calculating an angle from the jump distance is explained in text as “calculated the angles within a track spanned by three consecutive localizations⁵¹.” (lines 170-171) Providing a clarification sentence about how the measurement was done would be useful - was there a reference segment from which the angle was measured? How was it distinguished to measure either an acute or obtuse angle as an angle between three points could give you both?

We apologize for the misleading description of the angle analysis. We now added an illustration in Fig. 3e that defines the reference segment and the direction in which the angle is measured. We furthermore added the section “Jump angle analysis” to the Materials and Methods section in line 594-608:

“Jump angles were calculated as described previously⁴⁷. The angle θ spanned by two jumps (three consecutive localizations) indicates the change in direction of a molecule after it has moved in one direction (see inset in Figure 3e). While an angle of 0° indicates no change, i.e. a direct forward movement of the molecule, an angle of 180° indicates that a molecule moved backward in the opposite direction. Angles $<180^\circ$ or $>180^\circ$ indicate a movement to the right or left, respectively. The degree of reverse motion was quantified by calculating the ‘fold anisotropy’ metric, $f_{180/0}$ ⁶⁰, which measures how many-fold more likely a backward jump is, compared to a jump forward, using:

$$f(180/0) = \frac{BWD}{FWD} = P\left(\frac{180^\circ \pm 30^\circ}{0^\circ \pm 30^\circ}\right)$$

where BWD is the probability for a backward jump with an angle θ of 150° - 210° and FWD is the probability for a forward jump with an angle θ of 330° - 30° .

To calculate the angle between a jump and the region border, we first identified the point of the polygonal ROI that was closest to both detections of a jump. We then calculated the average of the orientation of the two neighboring polygonal border segments. Subsequently we determined the acute angle between the track segment and the ROI border segment.”

GRID as the acronym is first mentioned in line 201. The authors should briefly describe the method used for measuring survival time distributions.

We added the full form of the GRID acronym (Genuine rate identification) at the suggested position (line 215) and added the following description in the Materials and Methods section (line 633-636):

“In brief, GRID uses a superposition of 40 exponential functions with fixed decay rate constants between $10^{-2} s^{-1}$ and $10^2 s^{-1}$ and an appropriate set of regularizations to ensure robust convergence of the fit to the survival time distributions. As a result, GRID yields the amplitudes corresponding to each of the fixed decay rates.”

Why is the three component model (as opposed to two or one) is the most reasonable fit could be explained better. Obviously more parameters gives a better fit but what is the justification?

The reviewer legitimately points out that our comparison of fit-quality by the sum of squared errors (SSE) measure is not suited. A similar critic was also raised by Reviewer #2. We now use both the reduced χ^2 as well as the Akaike Information Criterion to compare the two- and the three-component models (Material and Methods section line 579-585 and Supplementary Fig. 3).

We also missed to explain why this is a reasonable model and to give an interpretation of each of the components. In many studies, the diffusive motion of molecules was found to be anomalous. In our experiments, anomalous diffusion likely results from diffusion in the limited space of the extracellular matrix and/or obstacles in this highly complex and

heterogeneous environment. We used a model of three diffusive populations, each representing ideal Brownian motion with distinct diffusion coefficients. We retrieved a very slow diffusion coefficient ($< 0.7 \mu\text{m}^2/\text{s}$) that we attributed to bound or immobile molecules, and two components with diffusion coefficients of $1\text{-}8 \mu\text{m}^2/\text{s}$ and $16\text{-}40 \mu\text{m}^2/\text{s}$, which together approximate anomalous diffusion. We added a corresponding paragraph in the text (line 151-156).

What were the criteria for manual identification of the cavities? Was there a width/size threshold? The top cavity in the first panel in figure 2a has a cavity region that is different in shape from the other cavities shown.

We defined cavities as regions, where the cell membranes of more than two cells meet, and where the memGFP signal of the involved cells are separated to a degree such that they are distinguishable. In contrast, we defined interfaces as region where the memGFP signal of two neighboring directly aligned cells are overlapping and cannot be distinguished due to the diffraction-limited resolution of the bright-field fluorescence image. We added this description in line 552-555.

The cavity region in Fig. 2a shows a different shape from the other cavities in this figure, because the two neighboring cavities appear to be connected to a certain degree. In this case, the memGFP signals of the membranes of the two cells touching the cavities are not overlapping.

Figure 6 d (ii), on the leftmost panel (0pg) there is still quite a bit of signal (possibly autofluorescence?). Please address.

The signal in all three panels originates from GFP protein that was injected into the extracellular space of the embryo at a late stage. The 0 pg refers to the amount of mRNA encoding the anti-GFP-nanobody that was injected at the one-cell stage and which itself is not fluorescing.

We acknowledge that our description may have been misleading and therefore replaced “GFP-nanobody” with “Anti-GFP-Nanobody” in Fig. 6d and added a label to the colorbar (“GFP fluorescence (a.u.)”).

Concerns - please discuss the limitations of the study regarding the following –

Validation of the Halo Tag attached versions of these proteins, whether they would reflect endogenous behavior and function of these morphogens is incomplete. In supplemental figure 1, authors show that the defects from injection with Sqt-HaloTag are similar to those from injection with Sqt-GFP. Please show any assays done to validate Cyc-HaloTag, Lefty1-HaloTag, and Lefty2-HaloTag. A better control would be to show that the range of action of the fusions is equivalent to the untagged counterparts since this would control for morphogen dispersal rather than just activity. Measuring expression of downstream genes (such as *gsc* or other nodal responsive genes) following mosaic expression of HaloTag tagged versions and untagged mRNA could be an additional approach. Some of the interpretations of the data such as finding more morphogen-HaloTag in cavities than in interfaces and rarely observing single molecules entering the cytoplasm (line 343-344) rely on the aspects of the HaloTag tagged version of the protein not validated here, such as how or whether they get cleared, endocytosed, degraded etc.

We agree with the reviewer that having proper controls of fluorescent fusion constructs is important. Therefore, over the last 10 years, we have already controlled for the dispersal and activity of the numerous fluorescent fusions of TGF- β superfamily ligands that we created. We demonstrated that the fluorescent fusions were similar to their untagged counterparts in

terms of processing, activity and signaling range (Müller et al. *Science* 2012, Almuedo-Castillo et al. *Nature Cell Biology* 2018, Soh et al. *Cell Reports* 2020, Pomreinke et al. *eLife* 2017). For convenience, below we briefly summarize the most important evidence supporting the proper activity and range of fluorescently tagged Nodals and Leftys before we describe how we quantitatively validated the HaloTag fusion proteins that we used in the present work.

Previously executed immunoblots of extracellular fractions showed that GFP- and Dendra2-tagged Nodals and Leftys are secreted, properly processed into mature ligands and present at levels similar to untagged ligands:

[redacted]

Quantitative PCR and *in situ* hybridization experiments showed that Nodal target genes are induced to a similar extent for tagged and untagged Nodals:

[redacted]

Phenotypic characterization, quantitative PCR and *in situ* hybridization experiments showed that tagged Leftys also have good activity:

[redacted]

Transplantation experiments demonstrated that tagged Nodals and tagged Leftys have activity ranges comparable to their untagged counterparts:

[redacted]

Strikingly, several tagged Nodal ligands can induce the formation of a full secondary axis, demonstrating that these fusion proteins have full biological activity to orchestrate complex morphogenetic processes. We also measured the gradients and signaling ranges of Squint-GFP and Squint-mVenus and found that they induce the key Nodal downstream effector pSmad2 with the same spatiotemporal dynamics as untagged Squint in the secondary axis formation assay:

[redacted]

Importantly, Lefty1-GFP forms relevant long-range gradients that can even rescue the strong phenotypic defects observed in maternal-zygotic *Lefty1;Lefty2* homozygous double mutants.

[redacted]

Our efforts from the last 10 years thus show that multiple tagged Nodal and Lefty constructs resulting from fusions with diverse fluorescent proteins such as GFP, Dendra2, mCherry and mVenus have biological activity and relevant signaling ranges that are comparable to their untagged counterparts. Given our previous efforts, the similarity in size between the HaloTag and GFP, as well as the inert behavior of the HaloTag seen in our

diffusion analysis, we reason that the dispersal of HaloTag fusions will not deviate from the dispersal of the other tags.

To quantitatively validate the activity of our HaloTag fusions, we have now performed additional qRT-PCR assays for the Nodal target gene *gsc*, similar to previously established assays for GFP fusions of Nodal and Lefty (Müller et al., *Science* (2012)). Since the HaloTag is similar in size to GFP, we used the previously validated GFP fusions (Müller et al., *Science* (2012)) as positive controls for proper activity.

We found that the HaloTag fusion proteins changed the expression levels of the Nodal target gene *gsc* similar to their GFP-tagged counterparts and have added these new results to Supplementary Fig. 1:

HaloTag fused to something secreted that is similar in size to Nodal and Lefty (e.g. GFP) would be an appropriate control for comparing the diffusion behavior.

We agree that the HaloTag fused to something that is similar in size to Nodal and Lefty is well suited to compare the diffusive behavior and to control our analysis pipeline. We therefore executed an additional experiment, where we measured sec-Halo-GFP. As expected, the diffusion coefficients of the sec-Halo-GFP were lower than those of the secreted HaloTag alone and similar to those of the morphogens. We added this information in line 166-168 and in Supplementary Table 1 and Supplementary Table 2, and we show the comparison between sec-Halo and sec-Halo-GFP in Supplementary Fig. 4:

Citations –

Should cite the following paper which shows a similar approach: Greiss, F., Deligiannaki, M., Jung, C., Gaul, U., & Braun, D. (2016). Single-Molecule Imaging in Living *Drosophila* Embryos with Reflected Light-Sheet Microscopy. *Biophysical journal*, 110(4), 939–946. <https://doi.org/10.1016/j.bpj.2015.12.035>

As suggested, we added this reference.

Reviewer #2:

In this excellent study, the authors apply single-molecule imaging and tracking in an *in vivo* context to directly study how the geometry and binding landscape in the extracellular environment of a developing tissue affects the effective diffusivities of different morphogens. They quantitatively characterize diffusive microstates of single morphogen molecules and disentangle the roles of surface receptor binding and geometry through well-conceived follow-up experiments and a simple but helpful computational model. Taken together, their work supports the hindered diffusion model of differential diffusivity, which marks an important contribution to ongoing efforts in developmental biology aimed at understanding the mechanisms of both Positional Information and Turing-style self-organized pattern formation.

The experimental results are well-presented and carefully interpreted, and the manuscript overall is clearly and concisely written and highly polished. Its publication will help popularize single-molecule imaging in cell and developmental biology, where such techniques are currently widely considered inaccessible. It will thus be of interest to a broad range of biologists for both its methodological advances and biological findings.

Given the above, I fully support the article's speedy publication.

We thank the reviewer for the positive assessment of our manuscript.

Comments on content

- The inferred diffusion coefficients should be compared to those derived from traditional methods, such as FCS. The authors argue that the results are consistent with the literature insofar that the different morphogens are very similar to each other in terms of their free diffusivity, but the actual magnitudes of the coefficients are not being compared. More

specifically, it would be relevant to state whether the coefficients of the unbound fractions (D3) match what has been reported from FCS measurements on freely diffusing molecules.

As suggested by the reviewer, we added the diffusion coefficients reported from FCS in line 160-162.

- The light sheet is approximately 3 microns thick, which raises the question whether the analysis needs to account for molecular motion in the z-axis. This may not be necessary or may be addressed within the TrackIt software, but either way the authors should clarify this point.

The reviewer is correct that the molecular motion in z-direction will bias the analysis. Bias arises due to the projection of 3D tracks onto a 2D plane, which underestimates diffusion coefficients, and from molecules moving out of the depth of focus of the objective (approximately 0.7 μm), which overestimates the fraction of bound molecules. We did not correct for these effects in our analysis. However, since all morphogens/conditions will experience similar bias, the comparisons between species we discuss will not be affected. We now state this in the text (line 141-142 and line 147-149) and in the Materials and Methods sections (line 570-575).

- Regarding the diffusion model fitting in Fig 3 and Supp Fig 4: The fact that the SSEs are lower in the 3-component model may be trivial given the model's additional degrees of freedom compared to the 2-component model. It would be more convincing if a measure was used that corrects for the number of parameters being estimated, such as the Akaike Information Criterion (AIC). Furthermore, the text does not make entirely clear what the authors' interpretation is for each of the three components (D1=bound, D2=?, D3=free).

The reviewer legitimately points out that our comparison of fit-quality by the sum of squared errors (SSE) measure is not suited. A similar critic was also raised by Reviewer #1. We now use both the reduced χ^2 as well as the Akaike Information Criterion suggested by this reviewer to compare the two- and the three-component models (Supplementary Fig. 3) and added a description in the Materials and Methods section (line 579-585).

We also missed to explain why this is a reasonable model and to give an interpretation of each of the components. In many studies the diffusive motion of molecules was found to be anomalous. In our experiment, anomalous diffusion likely results from diffusion in the limited space of the extracellular matrix and/or obstacles in this highly complex and heterogeneous environment. We used a model of three diffusive populations, each representing ideal Brownian motion with distinct diffusion coefficient. We retrieved a very slow diffusion coefficient ($< 0.7 \mu\text{m}^2/\text{s}$), which we attributed to bound or immobile molecules, and two components with diffusion coefficients of 1-8 $\mu\text{m}^2/\text{s}$ and 16-40 $\mu\text{m}^2/\text{s}$, which together approximate anomalous diffusion. We added a corresponding paragraph in the text in line 151-156.

- Regarding the angles (Fig 3e), it would be interesting to show how they relate to the angle of the relevant cell-cell interface (or the nearest boundary for cavities). Presumably, constrained diffusion would lead to more frequent reverse motion for angles more orthogonal to the boundary than for angles more parallel to the boundary. This would strengthen the authors' interpretation of the prevalence of reverse motion.

We thank the reviewer for this suggestion. We calculated the orientation of track segments with respect to the nearest region border (Supplementary Fig. 5b) and confirmed that tracks are mostly oriented alongside the membrane in interface regions and with more equally distributed orientations in cavities.

Next, we plotted the orientation against the jump angle (Supplementary Fig. 5c). As presumed by the reviewer, we found that forward motion ($\theta = 0^\circ$) mainly occurs for tracks oriented alongside region borders ($\alpha = 0^\circ$). For orientations more orthogonal to the boundary ($\alpha > 45^\circ$), the change in direction of the subsequent track segment was more drastic ($90^\circ < \theta < 270^\circ$). This is especially apparent in interface regions. In cavities, this effect is less pronounced, as particles that are further away from boundaries can diffuse freely without restrictions. We added a description on how the orientation towards region borders were calculated in the Materials and Methods section (line 605-608).

- In the methods, please state the thickness of the glass-bottom dishes used, as this is important for the optics.

As suggested by the reviewer, we added the thickness of the glass-bottom dish in the Material and Methods section.

Comments on presentation

- Fig 4c and Supp Fig 6b: It would be easier to read these figures if the y-axes were labeled with percentages (1%, 10%, 100%).

We changed the figure accordingly.

- Fig 6 and Supp Fig 9: The colormaps used are not perceptually uniform, which is no longer considered Good Practice in data visualization and should be changed. For the localization density plots in particular, the reader's perception of the patterns would be more accurate with a perceptually uniform colormap.

We agree that perceptually uniform colormaps as suggested by the reviewer are better suited. We therefore changed the colormaps in Fig. 1d ii), Fig. 2a, Fig. 6, Supplementary Fig. 9 and in Supplementary Movies 1-5 to the perceptually uniform colormap "Viridis".

- Fig 6b: This 3d figure is not easy to read, but I acknowledge that there is no real solution for this problem. Consider showing three 2d plots instead, one for each level of sigma (same ordering as 6c).

We agree that the 3D figure may need some time to fully capture the content. However, we think that more information can be extracted from this visualization, as the receptor densities sigma can also be compared directly. Nonetheless, we added three 2D plots, as suggested by the reviewer, to Supplementary Fig. 9.

Details

- Line 157: Use of the term "significant" should be avoided absent a statistical test for significance. Note that the resampling approach used to generate the s.d. in Fig 3 (and elsewhere) can also be used to perform hypothesis testing (see e.g. the tidymodels package "infer" for R), which would be applicable here. As the results are very clear, rewording would also be sufficient in this case.

As suggested, we replaced “significant” with “considerably”.

- Line 209: "remarkable agreement" seems to be an overstatement, given the differences in inferred and predicted Squint binding times (11s vs 4s).

As suggested, we replaced the wording and now use "good agreement" instead of “remarkable agreement”.

- Typo in Supp Fig 8: "a)" is repeated

We corrected this error.

- Typo in Line 328: "where" should be "were"

We corrected the typo.

REVIEWERS' COMMENTS

Reviewer #1 (Remarks to the Author):

The authors have adequately addressed all my concerns resulting in a fair, well-balanced paper. Job well done!

Reviewer #2 (Remarks to the Author):

The authors have satisfactorily addressed my comments. The analysis of jump angles relative to the ROI border in particular is a nice addition.

There are two very minor oversights that still need to be fixed:

- Supp. Figure 6a (formerly 6b): The y-axis labels should be changed to percentages. The authors have made this requested change in figure 4c, but must have missed that the same change should also be made in supplementary figure 6a.

- Line 169 (formerly 157): Contrary to the authors' statement in their response, "significantly" has not been changed to "considerably". Perhaps this change was made but then accidentally reverted.

These points notwithstanding, I'd like to commend the authors again for their thorough work and hope to see the manuscript published soon.

Point-by-point response to the reviewers' comments

Reviewer #1 (Remarks to the Author):

The authors have adequately addressed all my concerns resulting in a fair, well-balanced paper. Job well done!

We thank the reviewer for their fair assessment of our manuscript.

Reviewer #2 (Remarks to the Author):

The authors have satisfactorily addressed my comments. The analysis of jump angles relative to the ROI border in particular is a nice addition.

There are two very minor oversights that still need to be fixed:

- Supp. Figure 6a (formerly 6b): The y-axis labels should be changed to percentages. The authors have made this requested change in figure 4c, but must have missed that the same change should also be made in supplementary figure 6a.

We have made the change accordingly.

- Line 169 (formerly 157): Contrary to the authors' statement in their response, "significantly" has not been changed to "considerably". Perhaps this change was made but then accidentally reverted.

We have reintroduced the change accordingly.

These points notwithstanding, I'd like to commend the authors again for their thorough work and hope to see the manuscript published soon.

We thank the reviewer for their fair assessment of our manuscript.